REGISTERED REPORT

# Registered report: A chromatin-mediated reversible drug-tolerant state in cancer cell subpopulations

Babette Haven[1], Elysia Heilig[1], Cristine Donham[1], Michael Settles[1], Nicole Vasilevsky[2], Katherine Owen[3], Reproducibility Project: Cancer Biology*

[1]TGA Sciences, Medford, United States; [2]Oregon Health and Science University, Portland, United States; [3]University of Virginia, Charlottesville, United States

**Abstract** The Reproducibility Project: Cancer Biology seeks to address growing concerns about reproducibility in scientific research by conducting replications of selected experiments from a substantial number of high-profile papers in the field of cancer biology. The papers, which were published between 2010 and 2012, were selected on the basis of citations and Altmetric scores (Errington et al., 2014). This Registered Report describes the proposed replication plan of experiments from "A chromatin-mediated reversible drug-tolerant state in cancer cell subpopulations" by Sharma and colleagues, published in *Cell* in 2010 (Sharma et al., 2010). Sharma and colleagues demonstrated that prolonged exposure of cancer cells to TKIs give rise to small populations of "drug tolerant persisters" (DTPs) (Figure 1B-C) that were reversed during subsequent maintenance under drug-free conditions (Figures 1E, 2B and 2E). DTPs exhibited reduced histone acetylation and sensitivity to HDAC inhibitors (HDIs) (Figure 4A-B). Drug sensitivity was restored with co-treatment of either HDIs or an IGF-1R inhibitor, in combination with TKIs (Figure 5A-B). Inhibition of IGF-1R activation also led to decreased KDM5A expression and restoration of H3K4 methylation, suggesting a direct link between the IGF-1R signaling pathway and KDM5A function (Figure 7A, 7C, and 7I). The Reproducibility Project: Cancer Biology is a collaboration between the Center for Open Science and Science Exchange and the results of the replications will be published in *eLife*.

*For correspondence: tim@cos.io

**Group author details:**
Reproducibility Project: Cancer Biology See page 35

## Introduction

The effectiveness of chemotherapy for the treatment of cancer is limited by the acquisition of drug resistance. Intrinsic resistance, due to existing genetic alterations, as well as acquired resistance arising during treatment, can promote cross-resistance to structurally and functionally diverse drugs (*Kartal-Yandim et al., 2015*). Numerous mechanisms are involved in the development of multidrug resistance, including the overexpression of efflux proteins, decreased agent uptake, alterations in drug targets and modification of cell cycle checkpoints (*Engelman and Janne, 2008*; *Engelman and Settleman, 2008*; *Kartal-Yandim et al., 2015*; *Krishnamurty and Maly, 2010*). Moreover, it is increasingly recognized that tumors contain a high degree of molecular heterogeneity (*Greaves and Maley, 2012*), which can facilitate drug resistance through therapy-induced selection of resistant cell subpopulations present in the original tumor (*Swanton, 2012*).

In the developed world, non-small cell lung cancer (NSCLC) is the predominant form of the disease, accounting for 85% of lung cancer cases (*Allemani et al., 2015*; *Little et al., 2007*). Activation of the receptor tyrosine kinase epidermal growth factor receptor (EGFR) potentiates increased cell proliferation, migration, and survival (*Lynch et al., 2004*; *Paez et al., 2004*; *Pao et al., 2004*; *Sharma et al., 2007*). As a result, the *EGFR* gene has become an attractive target for small

molecular inhibitors. Tyrosine kinase inhibitors (TKIs) that target *EGFR*, such as gefitinib and erlotinib (*Pao et al., 2004*; *Sequist et al., 2008*; *Stella et al., 2012*), show initial clinical efficacy, however, the majority of patients eventually develop resistance to these chemotherapeutic agents (*Maemondo et al., 2010*; *Rosell et al., 2012*; *Stewart et al., 2015*). While a number of biological mechanisms of acquired resistance in NSCLC have been described (*Stewart et al., 2015*), in up to 30% of patients, the mechanism of resistance remains unknown (*Majem and Remon, 2013*). Sharma and colleagues have strongly implicated epigenetic changes as a key determinant in the maintenance of subpopulations of cancer cells with high-level drug resistance and potent tumorigenic capacity (*Sharma et al., 2007*).

In Figure 1B, the authors demonstrate that exposure of PC9 cells, a TKI-sensitive NSCLC cell line, to high concentrations of TKIs can select for a small subpopulation of "drug tolerant persister" cells (DTPs). The ability to generate PC9 persister cell populations in response to TKI treatment has been confirmed and extended by others (*Murakami et al., 2014*; *Ware et al., 2013*). Furthermore, the PC9 DTP phenotype was observed in a number of other TKI-sensitive cancer cells, including melanoma, colorectal, breast and gastric cancer-derived cell lines (*Sharma et al., 2007*). PC9 DTPs are largely quiescent, with the majority of cells remaining in G1 phase as determined by flow cytometric cell cycle analysis (Figure 1B). While not included in this replication attempt, prolonged exposure (30 + days) of DTPs to TKIs resulted in the generation of proliferating expanded persisters (DTEPs), as shown by colony outgrowth and cell cycle progression through S and G2/M phase (*Sharma et al., 2007*). Unlike parental PC9 cells, DTPs acquire cancer stem cell markers, such as CD133 (*Sharma et al., 2007*; Figure 2B) and CD24 (*Murakami et al., 2014*). Perhaps most importantly, TKI sensitivity was fully restored when PC9 DTPs were propagated in drug-free conditions (Figure 2E). These data indicate that acquired TKI resistance might not require permanent genetic alterations, an observation further supported by the fact that PC9 drug tolerance was not the result of genetic alterations resulting in enhanced drug efflux (Figure 1E). These results will be replicated in Protocols 2–5.

Deregulation of the epigenome is recognized as a common feature of many types of cancers. Multiple levels of epigenetic silencing have been defined in the alteration of gene expression, including DNA methylation and chromatin deacetylation (*Herman and Baylin, 2003*; *Shtivelman et al., 2014*). As shown in Figure 4A, PC9 DTPs exhibit reduced histone 3 acetylation compared to parental cells, and treatment of these cells with the histone deacetylase inhibitor (HDI) trichostatin A (TSA) resensitizes DTPs to apoptosis (Figure 4B). HDIs have been reported to induce a range of anticancer effects, such as tumor cell apoptosis, cell cycle arrest, differentiation, senescence, modulation of immune responses, and altered angiogenesis (*Bolden et al., 2006*). In Figure 5A,B, Sharma and colleagues tested the efficacy of 13 pharmacological anti-cancer agents, including 9 kinase inhibitors and 4 HDIs, in preventing the establishment of PC9 DTEPs. Inhibitors were used either as a single therapy or in combination with erlotinib, and colony outgrowth was assessed. Of the compounds examined, the combined treatment of HDIs (TSA, SAHA, scriptaid and MS275), together with erlotinib, prevented the emergence of DTEPs (Figure 5A,B). The synergistic interaction between TKIs and HDIs in reducing NSCLC viability has been described in several subsequent studies (*Chen et al., 2013*; *Kurtze et al., 2011*; *Nakagawa et al., 2013*). In addition to histone deacetylase inhibitors, the IGF-1R TKI AEW541 was also capable of inhibiting the emergence of DTEPs. Sharma and colleagues observed high basal expression of IGF binding protein 3 (IGFBP3), the high affinity binding partner of IGF-I and IGF-II, and phosphorylated IGF-1R in PC9 DTPs (Figure 7A). IGF-1R inhibition completely abolished IGF-1R activation (Figure 7C) and led to a significant reduction in the expression of KDM5A, a histone demethylase known to associate with histone deacetylases (Figure 7I), suggesting a direct link between the IGF-1R signaling pathway and KDM5A function. Work by Murakami and colleagues demonstrated a similar upregulation in IGF-1R activity in persistent TKI-selected NSCLC cells, although this phenotype was not attributed to chromatin modifications (*Murakami et al., 2014*). These results will be replicated in Protocols 4–9.

# Materials and methods

## Protocol 1: Preliminary study of growth characterisitics of PC9 cells and PC9 "drug-tolerant persisters" (DTPs)

This protocol describes the approach to determine growth characteristics of PC9 and PC9-derived DTPs (DTPs) prior to the initiation of the reproducibility of the experiments listed in the subsequent protocols. The experiments described will provide optimized parameters for subsequent protocols.

## Sampling

Each procedure will be performed once to access growth characteristics.

## Materials and reagents

| Reagent | Type | Manufacturer | Catalog # | Comments |
|---|---|---|---|---|
| PC9 Cells | Cell line | Sigma-Aldrich | 90071810 | Originally obtained from Dr. Kazuto Nishio (National Cancer Center Hospital, Tokyo) |
| 10 cm dishes | Labware | Corning | 430167 | Original brand not specified. |
| Erlotinib | Inhibitor | Cayman Chemical | 10483 | Replaces original obtained from MGH pharmacy |
| DMSO | Chemical | Sigma-Aldrich | D8418 | Not originally reported. |
| RPMI 1640 (4.5 g/l glucose) | Cell culture | ATCC | 30-2001 | Original brand not specified. |
| Fetal bovine serum (FBS) | Cell culture | HyClone | SH30910.03 | Original brand not specified. |
| Penicillin-Streptomycin (5,000 U/ml) | Cell culture | Life Technologies | 15070-063 | Not originally reported. |
| Trypsin | Cell culture | Life Technologies | 25200-056 | Not originally reported. |
| Accumax | Cell culture | Innovative Cell Technologies | AM-105 | Not originally reported. |
| Cell titer Glo | Cell viability assay | Promega | G7571 | Not originally included. |
| 96 well plates | Labware | Corning | 3997, 3610 or 3917 | Catalog # used depends on step of protocol |

## Procedure

### Note

- PC9 cells are grown in RPMI supplemented with 4.5 g/l glucose, 5% FBS and 1% penicillin/streptomycin at 37°C in a humidified atmosphere at 5% $CO_2$.
- Cells will be sent for mycoplasma testing and STR profiling.

1. Culture method:
   a. Thaw vial of PC9 cells in a 37°C water bath for 2–2.5 min. Transfer cells to a 15 ml conical tube containing 9 ml of pre-warmed growth medium.
   b. Pellet cells by centrifugation for 5 min at 100 x $g$. Aspirate the supernatant and resuspend the cells in 1 ml of pre-warmed growth medium.
   c. Determine the cell density and percent viability using an automated cell counter. Seed cells in T25, T75 or T175 cell culture flasks at a density of 20,000 cells/cm$^2$.
   d. Incubate the cells in a 37°C humidified incubator with 5% ambient $CO_2$ for at least 2 passages after thaw prior to using them for further experiments. Subculture cells as necessary to obtain the required number of cells for the procedures.
2. Determining the doubling time of PC9 cells:

   a. Seed PC9 cells at 2,000 cells/cm$^2$ in 10 cm dishes in 10 ml of growth medium/dish. Prepare 12 dishes. Record the date and time of cell addition to the dishes. Incubate in a 37°C humidified incubator with 5% ambient $CO_2$.
   b. Every 24 hr after addition to the dishes, detach cells from 1 dish using 0.25% trypsin/EDTA and determine the cell density and percent viability using an automated cell counter.
   c. Generate a growth chart by plotting the number of live cells/flask verse the time in culture.

3. Assessing DMSO solvent toxicity on PC9 cells:
   a. Seed PC9 cells into 96 well plates:
      i. Seed at 2500 cells/well in growth medium and incubate overnight to model the seeding density of cells during erlotinib IC$_{50}$ determination (Protocol 3).
      ii. Seed at 5800 cells/well in growth medium and incubate overnight to model the seeding density of cells during DTP generation (Protocol 2).
   b. Dilute 100% DMSO to 0.0005-2% in growth medium.
   c. Aspirate growth medium from the plate containing PC9 cells and add 60 µl of each DMSO dilution to the appropriate wells of the plate in triplicate. Add growth medium to the untreated control wells.
   d. Incubate the cells in the presence of DMSO dilutions for 3 days.
   e. Aspirate growth medium and add 100 µl/well of fresh growth medium to all wells containing cells and assay blank wells. Add 100 µl of CellTiter Glo to all wells containing growth medium. Measure luminescence using the Synergy 2 multi-mode plate reader from Bio-Tek.
   f. Subtract the assay blank value from all wells. Calculate the average RLU values of triplicate wells. Calculate the percent change in RLU values of the DMSO-treated wells from the untreated wells. Dilutions of DMSO that produce RLUs less than 95% of untreated will not be used for cell treatment in future experiments.

4. Determining the percentage of erlotinib-treated PC9 cells that become DTPs and the tolerance of DTPs and drug-withdrawn DTPs to trypsin or other detachment methods:
   a. Plate PC9 cells at 10$^6$ cells/dish (1.82 x 10$^4$ cells/cm$^2$) in a 10 cm tissue culture dish with grids in 10 ml/dish of complete growth medium. Prepare 16 x 10 cm dishes. Incubate cells for 24 hr in a 37°C/5% $CO_2$ incubator to allow cells to attach.
   b. Prepare erlotinib stock solution at the appropriate concentration in DMSO. If the results of step 3 allow, prepare a 60 mM erlotinib stock solution by dissolving 23.6 mg of erlotinib in 1 ml of DMSO. Prepare 60 µl aliquots and store at -20°C.
   c. Prepare erlotinib working dilution by adding the appropriate volume of erlotinib stock solution to 500 ml of growth medium for the appropriate final percentage of DMSO and so erlotinib is at a final concentration of 2 µM.
   d. Aspirate the growth medium from dishes, and replace growth medium with 10 ml/dish of erlotinib working dilution.
   e. Treat cells with erlotinib for 9 days, adding fresh medium and drug every 72 hr.
   f. At the end of 9 days of treatment, remove growth medium from the dishes and replace with 10 ml/dish of fresh growth medium without erlotinib. The viable cells should remain attached to the plate and are considered DTPs. For 8 plates proceed directly to detachment methods (step g), while with the other 8 plates propagate PC9 DTPs for multiple doublings in growth medium without erlotinib and then proceed to detachment methods (step g).
   g. Use the following methods to detach cells from 2 dishes each:
      i. 0.25% trypsin/EDTA at 37°C for $\leq$ 10 min.
      ii. 0.25% trypsin/EDTA at room temperature for $\leq$ 10 min.
      iii. 0.025% trypsin/EDTA at 2–8°C for $\leq$ 10 min.
      iv. Accumax at room temperature for 5–30 min.
   h. When cells have detached from the dishes, neutralize detachment enzymes as required, pool duplicates and pellet cell samples by centrifugation for 5 min at 100 x *g*.
   i. Aspirate supernatants and resuspend cells in 35 µl of pre-warmed growth medium. Determine cell density and viability using an automated cell counter and staining with trypan blue. The method with the highest yield and viability will be utilized.
      i. The confluency of the cells during the course of generating DTPs will be evaluated and this procedure might be repeated with a higher seeding density in an effort to maximize the number of DTPs generated. The cells should always be sub-confluent throughout the treatment period.

j. For drug-withdrawn PC9 DTPs, transfer 20 µl of each cell sample to one well of a 96-well plate containing 80 µl of growth medium. Label the wells with the detachment method and live cell by measuring staining with trypan blue. Incubate for 24 hr in a 37°C/5% $CO_2$ incubator. After 24 hr, check for cell attachment. Select the detachment method that has the least effect on cell viability, as measured by staining with trypan blue immediately after detachment, and that allows for cell recovery and growth after seeding into a new plate.

 i. If the number of cells in each dish is so low that it seems likely there will be too few cells to count with an automated cell counter, wait until drug-withdrawn DTPs begin to proliferate to perform this experiment.

5. Determining the doubling time of drug-withdrawn DTPs:
   a. Generate PC9 DTPs as described in step 4.
   b. Incubate DTPs in growth medium, without erlotinib, in a 37°C/5% $CO_2$ incubator, changing growth medium every 3 days.
   c. Using the detachment method selected in step 4, dissociate cells from 2 dishes every 24–48 hr and determine the cell count and viability of each cell sample using an automated cell counter and staining with trypan blue.
   d. Dissociate cells from 2 plates every 24–48 hr until a complete growth curve is established.

6. Assessing DMSO solvent toxicity on drug-withdrawn DTPs:
   a. Generate drug-withdrawn PC9 DTPs as described in step 4.
   b. After determining the best detachment method in step 4, use this detachment method to remove drug-withdrawn DTPs from dishes.
   c. Seed DTPs into the appropriate wells of a 96-well plate at 2,500 cells/well in growth medium. Incubate the plate overnight in a 37°C/5% $CO_2$ incubator to allow cells to attach.
   d. Dilute 100% DMSO to 0.0005-2% in growth medium.
   e. Aspirate growth medium from the plate containing drug-withdrawn PC9 DTPs and add 60 µl of each DMSO dilution to the appropriate wells of the plate in triplicate. Add growth medium to the untreated control wells.
   f. Incubate the cells in the presence of DMSO dilutions for 3 days.
   g. Aspirate growth medium and add 100 µl/well of fresh growth medium to all wells containing cells and assay blank wells. Add 100 µl of CellTiter Glo to all wells containing growth medium. Measure luminescence using the Synergy 2 multi-mode plate reader from Bio-Tek.
   h. Calculate the average values of triplicate wells. Subtract the assay blank value from all wells. Calculate the percent change in viability of the DMSO-treated wells from the untreated wells. Dilutions of DMSO that produce RLUs less than 95% of untreated will not be used for cell treatment in future experiments.

## Deliverables

- Data to be collected:
  - Step 2/Step 5: Raw data and growth chart of PC9 cells and drug-withdrawn DTPs.
  - Step 3/Step 6: Raw data and percent cell survival of CellTiter Glo assay.
  - Step 4: Raw counts and percent DTP generation.
  - Step 4: Cell density and viability data from detachment methods.

## Confirmatory analysis plan

○ n/a

## Known differences from the original study

All known differences are listed in the materials and reagents section above with the originally used item listed in the comments section. All differences have the same capabilities as the original and are not expected to alter the experimental design.

## Provisions for quality control

All of the raw data, will be uploaded to the project page on the OSF (https://osf.io/xbign) and made publically available.

**Protocol 2: Generation of PC9 DTPs**

This protocol describes the generation of DTPs from PC9 cells treated with erlotinib. The percent of cells surviving drug selection is calculated to replicate Figure 1C. The DTPs generated in the protocol will subsequently be used in protocols 3, 4, 5, 7, 8, and 9.

## Sampling

Each procedure will be performed for the following protocols:

- Protocol 3
- Protocol 4
- Protocol 5
- Protocol 7
- Protocol 8
- Protocol 9

## Materials and reagents

| Reagent | Type | Manufacturer | Catalog # | Comments |
|---|---|---|---|---|
| PC9 Cells | Cell line | Sigma-Aldrich | 90071810 | Originally obtained from Dr. Kazuto Nishio (National Cancer Center Hospital, Tokyo) |
| 10 cm dishes | Labware | Corning | 430167 | Original brand not specified. |
| Erlotinib | Inhibitor | Cayman Chemical | 10483 | Replaces original obtained from MGH pharmacy |
| DMSO | Chemical | Sigma-Aldrich | D8418 | Not originally reported. |
| RPMI 1640 (4.5 g/l glucose) | Cell culture | ATCC | 30-2001 | Original brand not specified. |
| Fetal bovine serum (FBS) | Cell culture | HyClone | SH30910.03 | Original brand not specified. |
| Penicillin-Streptomycin (5,000 U/ml) | Cell culture | Life Technologies | 15070-063 | Not originally reported. |

## Procedure

### Note

- PC9 cells are grown in RPMI supplemented with 4.5 g/l glucose, 5% FBS and 1% penicillin/streptomycin at 37°C in a humidified atmosphere at 5% $CO_2$.
- Cells will be sent for mycoplasma testing and STR profiling.

1. Plate PC9 cells at $10^6$ in 10 cm tissue culture dishes in media.
   a. Allow cells to become adherent before beginning drug treatment.
   b. Number of plates is dependent on percentage of erlotinib-treated PC9 cells that become DTPs (protocol 1, step 4), and required number of DTPs in the assay used.
2. Treat cells with 2 µM erlotinib (dissolved in DMSO) for 9 days, adding fresh media and drug every 72 hr.
   a. Determine cell count with an automated cell counter before starting drug treatment to serve as a baseline for percent survival (based on time from plating and doubling time of PC9 cells (protocol 1, step 2).
   b. Stock concentration used will be determined from assessing DMSO solvent toxicity on PC9 cells (protocol 1, step 3) to determine final DMSO concentration.
3. At the end of 9 days of treatment, the viable cells should remain attached to the plate, and these are considered DTPs. At the appropriate point indicated in each protocol, count the

number of cells with an automated cell counter surviving after treatment to calculate the percentage of cells surviving from the original population.

 a. Count the number of cells surviving after treatment in each dish and calculate the percentage of cells surviving from the original population. Calculate the average number of DTPs/dish as well as the standard deviation.

### Deliverables

- Data to be collected:
  - Count of number of cells surviving 9 days of drug treatment.
  - Quantification of the percent surviving cells from original population. (Figure 1C)
- Sample delivered for further analysis:
  - Erlotinib-resistant PC9-derived DTPs for protocols 3, 4, 5, 7, 8, and 9.

### Confirmatory analysis plan

- Meta-analysis of effect sizes:
  - Compare the replication data (mean percent PC9-derived DTPs and 95% confidence interval) to the original data (mean percent PC9-derived DTPs and 95% confidence interval) and use a random effects meta-analytic approach to combine the original and replication effects, which will be presented as a forest plot.

### Known differences from the original study

The replication attempt will be restricted to only the PC9 cell line since this is the line utilized in the other experiments of this replication attempt. All known differences are listed in the materials and reagents section above with the originally used item listed in the comments section. All differences have the same capabilities as the original and are not expected to alter the experimental design.

### Provisions for quality control

All of the raw data, will be uploaded to the project page on the OSF (https://osf.io/xbign) and made publically available.

## Protocol 3: Survival assay to determine the reversibility of DTP drug tolerance

This protocol assesses the sensitivity of PC9-derived DTPs to erlotinib following prolonged drug withdrawal. Erlotinib-resistant DTPs were generated as described in protocol 2, and then were cultured in the absence of drug for nine doublings. They were then exposed to a range of erlotinib concentrations (~0–2 µM) for 72 hr and survival was assessed. Drug-naïve PC9 cells were used as a control. This protocol serves to replicate data described in Figure 2E.

### Sampling

This experiment will be repeated 4 times.

- See Power Calculations section for details.

Experiment has 2 cohorts

- Cohort 1: Drug-naïve PC9 cells
- Cohort 2: Erlotinib-resistant PC9 DTPs cultured in drug-free medium for nine doublings

Each cohort has 6 conditions to be performed with four technical repeats per experiment:

- DMSO (vehicle)
- 0.0002 µM erlotinib
- 0.002 µM erlotinib
- 0.02 µM erlotinib
- 0.2 µM erlotinib
- 2 µM erlotinib

## Materials and reagents

| Reagent | Type | Manufacturer | Catalog # | Comments |
|---|---|---|---|---|
| PC9 Cells | Human cell line | Sigma-Aldrich | 90071810 | Originally obtained from Dr. Kazuto Nishio (National Cancer Center Hospital, Tokyo) |
| PC9 DTP cells | Human cell line | n/a | n/a | Generated according to protocol 1 |
| Erlotinib | Inhibitor | Cayman Chemical | 10483 | Replaces original obtained from MGH pharmacy |
| DMSO | Chemical | Sigma-Aldrich | D8418 | Not originally reported. |
| RPMI 1640 (4.5 g/l glucose) | Cell culture | ATCC | 30-2001 | Original brand not specified |
| Fetal bovine serum (FBS) | Cell culture | HyClone | SH30910.03 | Original brand not specified |
| Penicillin-Streptomycin (5,000 U/ml) | Cell culture | Life Technologies | 15070-063 | Not originally reported. |
| Phosphate buffered saline (PBS) | Buffer | Life Technologies | 14190 | Original brand not specified |
| Formaldehyde | Chemical | Fisher Scientific | F79-1 | Original brand not specified |
| Syto60 | Nucleic acid stain | Molecular Probes | S11342 | Original catalog number not specified |
| Odyssey Infrared Imager | Instrument | Li-Cor Biosciences | CLx | |
| Image Studio | Software | Li-Cor Biosciences | | |
| 96-well plates | Labware | Corning | 3603 | Replaces 12-well dishes originally used. |

## Procedure

### Note

- PC9 cells are grown in RPMI supplemented with 4.5 g/l glucose, 5% FBS and 1% penicillin/streptomycin at 37°C in a humidified atmosphere at 5% $CO_2$.
- Cells will be sent for mycoplasma testing and STR profiling.

1. Generate PC9-derived DTPs with 2 µM erlotinib in 10 cm tissue culture dishes, as described in protocol 2.
2. Replace with drug-free medium and propagate PC9 DTPs for 9 doublings.
   a. Doubling time of drug-withdrawn DTPs (protocol 1, step 5) will be used to determine length of time of this step.
3. Plate drug naïve PC9 or drug-withdrawn PC9 DTP cells at 2,500 cells/well in 96-well plates in quadruplicate for each concentration (0–10 µM) of drug to be used.
   a. The detachment method identified in protocol 1, step 4 will be used.
4. 24 hr after plating, replace medium with medium containing the indicated concentrations of erlotinib or DMSO.
   a. Stock concentration used will be determined from assessing DMSO solvent toxicity on PC9 cells (protocol 1, step 3) and drug-withdrawn DTPs (protocol 1, step 6) to determine final DMSO concentration.
5. 72 hr after drug treatment, remove medium and wash cells with phosphate buffered saline (PBS).
6. Fix cells for 15 min with 4% formaldehyde in PBS at room temperature.
7. Wash cells with PBS, 3 x 10 min.
8. Stain cells with 1 nM Syto60 in PBS for 15 min at room temperature following manufacturer's protocol.

9. Remove dye and wash cells with PBS.
10. Quantify fluorescence at 700 nm with an Odyssey Infared Imager following manufacturer's instructions.
11. Calculate the absolute $IC_{50}$ value for each cohort.
12. Repeat independently three additional times.

## Deliverables

- Data to be collected:
    - Fluorescence readings at 700 nm for each sample.
    - Quantification of percent survival relative to vehicle treated samples.
    - Graph of percent survival for each cell line relative to untreated cells plotted against each concentration of erlotinib used. (Compare to Figure 2E)
    - Absolute $IC_{50}$ values for each sample.

## Confirmatory analysis plan

- Statistical Analysis of the Replication Data:
    - Unpaired two-tailed *t*-test of the percent survival relative to vehicle for drug-naïve PC9 cells compared to erlotinib-resistant PC9 DTPs cultured in drug-free medium for nine doublings.
- Meta-analysis of original and replication attempt effect sizes:
    - Compute the effect sizes of each comparison, compare them against the effect size in the original paper and use a random effects meta-analytic approach to combine the original and replication effects, which will be presented as a forest plot.

## Known differences from the original study

The original study used 12-well dishes to assess survival. This replication attempt will use 96 well dishes using the same starting cell density as originally used to minimize the number of DTPs needed. All known differences are listed in the materials and reagents section above with the originally used item listed in the comments section. All differences have the same capabilities as the original and are not expected to alter the experimental design.

## Provisions for quality control

All of the raw data will be uploaded to the project page on the OSF (https://osf.io/xbign) and made publically available.

## Protocol 4: Western blot for histone H3K14 acetylation and CD133 expression in PC9 cells and PC9-derived DTPs

This protocol seeks to characterize phenotypic differences between native PC9 cells and PC9-derived DTPs. Immunoblotting will be used to compare levels of acetylated histone H3K4, as well as the expression of the cancer stem cell marker CD133, in both PC9 cells and PC9-derived DTPs. Additionally, this protocol will include measuring phosphorylated EGFR and total EGFR in both PC9 cells and PC9-derived DTPs as a measure of erlotinib efficacy. This protocol is a replication of Figure 2B, Figure 4A (upper panel) and Figure 1E (upper right panel).

## Sampling

This experiment will be repeated a total of 4 times.

The original data is qualitative, thus to determine an appropriate number of replicates to initially perform, sample sizes based on a range of potential variance was determined.

- See Power Calculations section for details.

Experiment has 2 cohorts:

- Cohort 1: Drug-naïve PC9 cells
- Cohort 2: Erlotinib-resistant PC9 DTPs

Western blotting is performed for the following proteins:

- Acetylated H3K14
- CD133

- pEGFR (Y1068)
- H3 (total histone protein control) [additional]
- EGFR (total EGFR protein control)
- GAPDH (control)

## Materials and reagents

| Reagent | Type | Manufacturer | Catalog # | Comments |
| --- | --- | --- | --- | --- |
| PC9 Cells | Human cell line | Sigma-Aldrich | 90071810 | Originally obtained from Dr. Kazuto Nishio (National Cancer Center Hospital, Tokyo) |
| PC9 DTP cells | Human cell line | n/a | n/a | Generated according to protocol 2 |
| 10 cm dishes | Labware | Corning | 430167 | Original brand not specified. |
| RPMI 1640 (4.5 g/l glucose) | Cell culture | ATCC | 30-2001 | Original brand not specified |
| Fetal bovine serum (FBS) | Cell culture | HyClone | SH30910.03 | Original brand not specified |
| Penicillin-Streptomycin (5,000 U/ml) | Cell culture | Life Technologies | 15070-063 | Not originally reported. |
| RIPA buffer | Buffer | Sigma-Aldrich | R0278 | From replicating lab protocol |
| Complete mini protease inhibitor cocktail tablets | Inhibitor | Roche Diagnostics | 11 836 153 001 | From replicating lab protocol |
| Halt phosphatase inhibitor cocktail | Inhibitor | Thermo Scientific | 1862495 | From replicating lab protocol |
| Phenylmethanesulfonyl fluoride solution | Inhibitor | Sigma-Aldrich | 93482 | |
| Pierce BCA Protein Assay Kit | Protein Assay | Thermo Fisher | 23225 | From replicating lab protocol |
| NuPAGE LDS sample buffer (4X) | Western blot reagent | Life Technologies | NP0007 | Replaces Laemmli sample buffer |
| Molecular weight markers | Western blot reagent | Li-Cor | 928-40000 | Not originally reported. |
| NuPage Sample Reducing Agent (10X) | Western blot reagent | Life Technologies | NP0004 | From replicating lab protocol |
| NuPAGE 4-12% Bis-Tris gels (10 well/15 well) | Western blot reagent | Life Technologies | NP0335BOX/ NP0336BOX | From replicating lab protocol |
| NuPAGE MES SDS Running Buffer (20X) | Western blot reagent | Life Technologies | NP0002 | From replicating lab protocol |
| NuPAGE MOPS SDS Running Buffer (20X) | Western blot reagent | Life Technologies | NP0001 | From replicating lab protocol |
| iBlot gel transfer stacks nitrocellulose | Western blot reagent | Life Technologies | IB301002 | From replicating lab protocol |
| Rabbit anti-H3K14Ac antibody | Antibodies | Active Motif | 39698 | Original catalog number not specified. Note that each new batch is associated with a new catalog number. |

*Continued on next page*

*Continued*

| Reagent | Type | Manufacturer | Catalog # | Comments |
|---------|------|--------------|-----------|----------|
| Rabbit anti-GAPDH antibody | Antibodies | Life Technologies | PA1-988 | Original was from Biosource International (acquired by Life Technologies) 1:500-1:5000 suggested dilution |
| Mouse anti-GAPDH antibody (clone GA1R) | Antibodies | Life Technologies | MA5-15738 | Original was from Biosource International (acquired by Life Technologies) 1:1000-1:10,000 suggested dilution |
| Mouse anti-H3 antibody (clone 1B1-B2) | Antibodies | Active Motif | 61476 | Original catalog number not specified. Note that each new batch is associated with a new catalog number. |
| Mouse anti-CD133 antibody (clone 17A6.1) | Antibodies | EMD Millipore | MAB4399 | Replaces Cell Signaling Technology brand (discontinued) |
| Rabbit anti-pEGFR (Y1068) antibody (clone D7A5) | Antibodies | Cell Signaling Technology | 3777 | Original catalog number not specified. |
| Mouse anti-EGFR antibody (clone 1F4) | Antibodies | Cell Signaling Technology | 2239 | Replaces Santa Cruz brand. |
| Donkey anti-mouse IRDye 680RD | Antibodies | Li-Cor | 926-68072 | Replaces HRP conjugated antibodies |
| Donkey anti-rabbit IRDye 800CW | Antibodies | Li-Cor | 926-32213 | |
| Odyssey Infrared Imager | Instrument | Li-Cor Biosciences | CLx | |
| Image Studio | Software | Li-Cor Biosciences | | |

## Procedure
### Note

- PC9 cells are grown in RPMI supplemented with 4.5 g/l glucose, 5% FBS and 1% penicillin/ streptomycin at 37°C in a humidified atmosphere at 5% $CO_2$.
- Cells will be sent for mycoplasma testing and STR profiling.

1. Generate PC9-derived DTPs with 2 μM erlotinib in 10 cm tissue culture dishes, as described in protocol 2.
2. Plate drug naïve PC9 cells in 10 cm tissue culture dishes 2 days prior to harvest at sparse density.
   a. Seed cells at a density that will allow sufficient cells for analysis while remaining sub-confluent.
3. Dissociate cells from plates and count. Harvest drug naïve PC9 and PC9-derived DTPs in complete lysis buffer following replicating labs standard procedure.
   a. The detachment method identified in protocol 1, step 4 will be used.
   b. Complete lysis buffer: RIPA lysis buffer supplemented with 1X phosphatase inhibitor, protease inhibitor cocktail, 1 mM PMSF.
4. Normalize gel loading to total cell number, add 4X LDS sample buffer supplemented with reducing agent, and denature at 70°C for 10 min.
   a. Determine protein concentration by BCA assay following manufacturer's instructions for lysates from drug-naïve PC9 cells to determine the total protein concentrations relative to total cell number. Since erlotinib-resistant PC9 DTPs are anticipated to be less abundant, the PC9 BCA results will be used to approximate the concentration of DTPs to ensure enough protein is loaded.
5. Separate equivalent number of cells (~10–60 μg of protein) per lane with protein ladder and transfer to a membrane using the replicating labs standard procedures.

6. After transfer, block non-specific binding and immunoblot membrane with the following combinations of primary antibodies at the dilution/concentration recommended by the supplier.
   a. Rabbit anti-H3K14Ac (17 kDa) at 0.5-2 µg/ml and mouse anti-H3 (17 kDa) at 0.25 µg/ml
   b. Rabbit anti-H3K14Ac (17 kDa) at 0.5-2 µg/ml and mouse anti-GAPDH (37 kDa) at 1:1000-1:10,000
   c. Mouse anti-CD133 (97 kDa) at 0.2 µg/ml and rabbit anti-GAPDH (37 kDa) at 1:500-1:5000
   d. Rabbit anti-pEGFR (Y1068) (175 kDa) at 1:1000 and mouse anti-EGFR (175 kDa) at 1:1000 dilution

**Protocol 4 Western Blot Antibody Multiplexing**

| | Protein of interest | | Loading control | |
|---|---|---|---|---|
| Combination | Description | Working Conc. | Description | Working Conc. |
| 1 | Rabbit anti-H3K14Ac (17 kDa) | 0.5–2 µg/ml | Mouse anti-H3 (17 kDa) | 0.25 µg/ml |
| 2 | Rabbit anti-H3K14Ac (17 kDa) | 0.5–2 µg/ml | Mouse anti-GAPDH (37 kDa) | 1:500–1:5000 |
| 3 | Mouse anti-CD133 (97 kDa) | 0.2 µg/ml | Rabbit anti-GAPDH (37 kDa) | 1:500–1:5000 |
| 4 | Rabbit anti-pEGFR (Y1068) (175 kDa) | 1:1000 | Mouse anti-EGFR (175 kDa) | 1:1000 |

7. Wash and apply appropriate secondary antibodies for 1 hr at RT with constant agitation and detect signal using Odyssey imaging system.
8. Analyze bands with Image Studio software and normalize to loading controls.
   a. H3K14Ac normalized to H3 (total) [additional]
   b. H3K14Ac normalized to GAPDH
   c. CD133 normalized to GAPDH
   d. pEGFR (Y1068) normalized to EGFR (total)
9. Repeat independently three additional times.

## Deliverables

- Data to be collected:
  - Full scans for each Western blot with ladder. (Compare to Figure 4A (upper panel), Figure 1E (upper right panel), and Figure 2B)
  - Raw data of band analysis and normalized bands for each sample.

## Confirmatory analysis plan

- Statistical Analysis of the Replication Data:
  - The following three tests will be performed using the Bonferroni correction because the Western blots are all from the same samples:
    - Unpaired two-tailed *t*-test of H3K14Ac levels normalized to GAPDH from drug-naïve PC9 cells compared to erlotinib-resistant PC9 DTPs.
    - Unpaired two-tailed *t*-test of normalized H3K14Ac levels from drug-naïve PC9 cells compared to erlotinib-resistant PC9 DTPs.
    - Unpaired two-tailed *t*-test of normalized pEGFR (Y1068) levels from drug-naïve PC9 cells compared to erlotinib-resistant PC9 DTP
- Meta-analysis of original and replication attempt effect sizes
  - The replication data (mean and 95% confidence interval) will be plotted with the original reported data value, where possible, plotted as a single point on the same plot for comparison.
- Additional exploratory analysis:
  - H3K14Ac levels will also be normalized to H3 (total) and the same analysis described above will be performed, which serves as an independent normalization control not included in the original report.

## Known differences from the original study

An additional control, total H3, was added to this replication attempt that was not originally reported in Figure 4A (top panel) to normalize the H3K14 acetylation levels. The original loading control of GAPDH will also be utilized to allow for a direct comparison. This replication attempt will prepare cells in RIPA lysis buffer, while it is unclear if the original study used a lysis buffer, lysed cells directly in Laemmli sample buffer, or used acid extraction for histones. All known differences are listed in the materials and reagents section above with the originally used item listed in the comments section. All differences have the same capabilities as the original and are not expected to alter the experimental design.

## Provisions for quality control

The samples will be normalized based on total cell number, similar to the original report, but protein concentration will also be determined to ensure enough lysate is loaded to ensure detection of the proteins of interest. All of the raw data will be uploaded to the project page on the OSF (https://osf.io/xbign) and made publically available.

## Protocol 5: Flow cytometry to analyze sensitivity of PC9 cells to HDAC inhibition

This protocol tests the sensitivity of PC9 cells or PC9-derived DTPs to the HDAC inhibitor Trichostatin A (TSA). Flow cytometry is utilized following BrdU staining to analyze the cell cycle distribution of cells treated with TSA. In particular, sub-G1 apoptotic cells are quantified. It is a replication of the experiment presented in Figure 4B (upper panel). The additional cell-cycle data collected in this protocol will also serve as a replication of Figure 1B (lower panel), as a means to characterize phenotypic differences between native PC9 cells and PC9-derived DTPs.

### Sampling

This experiment will be repeated a total of 3 times for a minimum power of 80%.

- See Power Calculations section for details.

Experiment has 2 cohorts

- Cohort 1: Drug-naïve PC9 cells
- Cohort 2: Erlotinib-resistant PC9 DTPs

Each cohort will be treated with the following

- DMSO (Vehicle control) [additional]
- untreated
- 50 nM TSA
- 100 nM TSA

Flow cytometry will be performed with the following antibodies or controls

- Anti-BrdU
- Unstained cells (negative control)
- No anti-BrdU (secondary only control)

### Materials and reagents

| Reagent | Type | Manufacturer | Catalog # | Comments |
|---------|------|--------------|-----------|----------|
| PC9 Cells | Human cell line | Sigma-Aldrich | 90071810 | Originally obtained from Dr. Kazuto Nishio (National Cancer Center Hospital, Tokyo) |

*Continued on next page*

*Continued*

| Reagent | Type | Manufacturer | Catalog # | Comments |
|---|---|---|---|---|
| PC9 DTP cells | Human cell line | n/a | n/a | Generated according to protocol 2 |
| 10 cm dishes | Labware | Corning | 430167 | Original brand not specified. |
| RPMI 1640 (4.5 g/l glucose) | Cell culture | ATCC | 30-2001 | Original brand not specified. |
| Fetal bovine serum (FBS) | Cell culture | HyClone | SH30910.03 | Original brand not specified. |
| Penicillin-Streptomycin (5,000 U/ml) | Cell culture | Life Technologies | 15070-063 | Not originally reported. |
| Trichostatin A (TSA) | Inhibitor | Enzo Life Sciences (Biomol) | BML-GR309-0001 | Original catalog number not specified |
| DMSO | Chemical | Sigma-Aldrich | D8418 | Not originally reported |
| Cell Labeling Reagent (BrdU) | | GE Healthcare | RPN201 | Original was from Amersham Pharmacia (acquired by GE Healthcare) |
| Anti-BrdU antibody (clone B44) | Antibodies | Becton-Dickinson | 347580 | Original catalog number not specified. |
| FITC-conjugated goat anti-mouse secondary antibody | Antibodies | Jackson Immunoreserach | 115-095-146 | Replaces Vector laboratories brand |
| 2 M HCl | Chemical | Sigma-Aldrich | 71826-1L | Original brand not specified. |
| 0.5% Triton X-100 | Chemical | Fisher Scientific | BP151-100 | Original brand not specified. |
| 0.1 M NaB4O7·10H2O (pH 8.5) | Chemical | Sigma-Aldrich | B9876 | Original brand not specified. |
| 5ug/ml propidium iodide | Chemical | BD Biosciences | 556463 | Replaces Sigma-Aldrich brand |
| RNAse A | Enzyme | Sigma-Aldrich | R6513 | Original catalog number not specified. |
| Ethanol | Chemical | Sigma-Aldrich | E7023 | Original brand not specified. |
| Flow cytometer | Instrument | Becton Dickinson | FACScan | |
| Cell Quest | Software | Becton Dickinson | | |

## Procedure

### Note

- PC9 cells are grown in RPMI supplemented with 4.5 g/l glucose, 5% FBS and 1% penicillin/streptomycin at 37°C in a humidified atmosphere at 5% $CO_2$.
- Cells will be sent for mycoplasma testing and STR profiling.

1. Generate PC9-derived DTPs with 2 µM erlotinib in 10 cm tissue culture dishes, as described in protocol 2.
2. Plate drug naïve PC9 cells in 10 cm tissue culture dishes 2 days prior to harvest at sparse density.
   a. Seed cells at a density that will allow sufficient cells for analysis while remaining sub-confluent.
3. Treat cells with DMSO (vehicle), 50 nM TSA, or 100 nM TSA, or leave untreated, for 20 hr.
   a. Stock concentration used will be determined from assessing DMSO solvent toxicity on PC9 cells (protocol 1, step 3) and drug-withdrawn DTPs (protocol 1, step 6) to determine final DMSO concentration.
4. Dissociate cells from plates and count.
   a. The detachment method identified in protocol 1, step 4 will be used.
   b. Include adherent and floating cells.
5. Incubate cells with Cell Labeling Reagent at 37°C for 1 hr following manufacturer's instructions.
   a. Include unstained cells (negative control)
   b. No anti-BrdU control (secondary only control)

6. Fix and stain cells with anti-BrdU following manufacturer's instructions with the following modifications:
    a. Wash all cells (adherent and floating) with PBS.
    b. Fix cells with 80% ethanol.
    c. Denature DNA for 30 min with 2 M HCl/0.5% Triton X-100.
    d. Neutralize DNA with 0.1 M $NaB_4O_7 \cdot 10H_2O$ (pH 8.5).
    e. Incubate cells with an anti-BrdU antibody at 20 µl per $10^6$ cells according to manufacturer's instructions.
    f. Incubate cells with a FITC-conjugated goat anti-mouse secondary antibody diluted 1:50.
7. Treat cells with RNase A following manufacturer's instructions.
8. Stain cells with 5ug/ml propidium iodide following manufacturer's instructions.
9. Analyze cells using two-dimensional FACS analysis and CELLQUEST software.
    a. Determine percentage of cells in the various phases of the cell cycle (G1, S, and G2M) including the sub-G1 population.
10. Repeat independently two additional times.

## Deliverables

- Data to be collected:
    - All flow populations including each gating step plus histogram plots for final population
    - Analysis of cell cycle distribution. (Compare to Figure 1B)
    - Bar graph depicting the percent sub-G1 population for each cohort. (Compare to Figure 4B)

## Confirmatory analysis plan

- Statistical Analysis of the Replication Data:
    - Cochran-Mantel-Haenszel test (3x2x3 contingency table) of the percent of cells in the different cell cycle phases (G1, S, and G2M) of drug-naïve PC9 cells compared to erlotinib-resistant PC9 DTPs cells left untreated, while controlling for the number of times the experiment is performed.
    - Two-way ANOVA of percent of sub-G1 cells from drug-naïve PC9 and erlotinib-resistant PC9 DTPs cells left untreated or treated with 50 nM TSA, or 100 nM TSA with the following planned comparisons using the Bonferroni correction:
        - Percent of sub-G1 cells from drug-naïve PC9 cells untreated compared to 50 nM TSA.
        - Percent of sub-G1 cells from drug-naïve PC9 cells untreated compared to 100 nM TSA.
        - Percent of sub-G1 cells from erlotinib-resistant PC9 DTPs untreated compared to 50 nM TSA.
        - Percent of sub-G1 cells from erlotinib-resistant PC9 DTPs untreated compared to 100 nM TSA.
- Meta-analysis of original and replication attempt effect sizes:
    - Compute the effect sizes of each comparison, compare them against the effect size in the original paper and use a random effects meta-analytic approach to combine the original and replication effects, which will be presented as a forest plot.

## Known differences from the original study

The replication attempt will be restricted to drug-naïve PC9 cells and PC9 DTPs since these are utilized in the other experiments of this replication attempt. The original analysis of the percent of cells in the different cell cycle phases (G1, S, and G2M) reported in Figure 1B used untreated cells, while this replication will also include vehicle (DMSO) treated cells. This is an additional exploratory measure to understand if DMSO treatment has an impact on the viability and cell cycle profile of the cells. The original paper reported the anti-BrdU antibody was diluted at 1:500, however since the antibody used in this replication attempt might not be the same, the manufacturer's instructions will be followed. All known differences are listed in the materials and reagents section above with the originally used item listed in the comments section. All differences have the same capabilities as the original and are not expected to alter the experimental design.

## Provisions for quality control

All of the raw data will be uploaded to the project page on the OSF (https://osf.io/xbign) and made publically available.

## Protocol 6: Preventing the establishment of drug-tolerant colonies

This protocol tests the ability of the HDAC inhibitor TSA and the IGF-1R kinase inhibitor AEW541 to prevent the formation of drug tolerant populations of PC9 cells. As a control, native PC9 cells are also treated with the inhibitors in the absence of erlotinib. It is a replication of Figure 5A and Figure 5B.

### Sampling

This experiment will be repeated a total of 3 times for a minimum power of 99%.

- See Power Calculations section for details.

Experiment has 6 conditions (all conditions use drug-naïve PC9 cells):

- Untreated (cultured for 6 days)
- Treated with 20 nM TSA (cultured for 6 days)
- Treated with 0.5 µM AEW541 (cultured for 6 days)
- Treated with 2 µM erlotinib (cultured for 33 days)
- Treated with 2 µM erlotinib + 20 nM TSA (cultured for 33 days)
- Treated with 2 µM erlotinib + 0.5 µM AEW541 (cultured for 33 days)

### Materials and reagents

| Reagent | Type | Manufacturer | Catalog # | Comments |
|---|---|---|---|---|
| PC9 Cells | Human cell line | Sigma-Aldrich | 90071810 | Originally obtained from Dr. Kazuto Nishio (National Cancer Center Hospital, Tokyo) |
| 10 cm dishes | Labware | Corning | 430167 | Original brand not specified. |
| Erlotinib | Inhibitor | Cayman Chemical | 10483 | Replaces original obtained from MGH pharmacy |
| DMSO | Chemical | Sigma-Aldrich | D8418 | Not originally reported. |
| RPMI 1640 (4.5 g/l glucose) | Cell culture | ATCC | 30-2001 | Original brand not specified. |
| Fetal bovine serum (FBS) | Cell culture | HyClone | SH30910.03 | Original brand not specified. |
| Penicillin-Streptomycin (5,000 U/ml) | Cell culture | Life Technologies | 15070-063 | Not originally reported. |
| Trichostatin A (TSA) | Inhibitor | Enzo Life Sciences (Biomol) | BML-GR309-0001 | Original catalog number not specified |
| AEW541 | Inhibitor | Cayman Chemical | 13641 | Replaces original obtained from Novartis Pharmaceuticals |
| Giemsa stain | Chemical | Ricca Chemical Company | 3250-4 | Original brand not specified |
| Methanol | Chemical | Specific brand information will be left up to the discretion of the replicating lab and recorded later | | Reagent needed from Giemsa staining protocol |
| Phosphate buffered saline (PBS) | Buffer | Life Technologies | 14190 | Original brand not specified |

### Procedure

Note

- PC9 cells are grown in RPMI supplemented with 4.5 g/l glucose, 5% FBS and 1% penicillin/streptomycin at 37°C in a humidified atmosphere at 5% $CO_2$.
- Cells will be sent for mycoplasma testing and STR profiling.

1. Plate PC9 cells at 1 x 10$^5$ in 10 cm tissue culture dishes in medium.
   a. Allow cells to become adherent before beginning drug treatment.
2. 24 hr after plating, treat cells with the pharmacological agents for the following times:
   a. Stock concentration used will be determined from assessing DMSO solvent toxicity on PC9 cells (protocol 1, step 3) to determine final DMSO concentration.
   b. Untreated - 6 days
   c. DMSO (vehicle) treated - 6 days
   d. 20 nM TSA - 6 days
   e. 0.5 µM AEW541 - 6 days
   f. 2 µM erlotinib - 33 days
   g. 2 µM erlotinib + 20 nM TSA - 33 days
   h. 2 µM erlotinib + 0.5 µM AEW541 - 33 days
3. Replace with fresh media containing relevant drugs every 72 hr.
4. After the appropriate length of drug treatment, remove media and fix cells with ice-cold methanol for 5 min at room temperature. Stain with Giemsa following manufacturer's instructions.
5. Blindly analyze culture dishes by light microscopy (40X magnification) and manually count the number of individual colonies present.
6. Repeat independently two additional times.

## Deliverables

- ▪ Data to be collected:
  - Images of plates after staining. (Compare to Figure 5A)
  - Bar graph of number of resistant colonies formed after each treatment. (Compare to Figure 5B)

## Confirmatory analysis plan

- Statistical Analysis of the Replication Data:
  - Wilcoxon-Mann Whitney test of colony numbers of drug-naïve PC9 cells treated with erlotinib, erlotinib + TSA, and erlotinib + AEW541 with the following comparisons using the Bonferroni correction:
    - Number of colonies from drug-naïve PC9 cells treated with erlotinib compared to erlotinib + TSA.
    - Number of colonies from drug-naïve PC9 cells treated with erlotinib compared to erlotinib + AEW541.
- Meta-analysis of original and replication attempt effect sizes:
  - Compute the effect sizes of each comparison, compare them against the effect size in the original paper and use a random effects meta-analytic approach to combine the original and replication effects, which will be presented as a forest plot.

## Known differences from the original study

The replication attempt will be restricted to TSA and AEW541 inhibitors and not include the other pharmacological agents originally reported since only these inhibitors are utilized in the other experiments of this replication attempt. The original study used formaldehyde to fix the cells, while this replication attempt will use methanol as recommended by the manufacturer. All known differences are listed in the materials and reagents section above with the originally used item listed in the comments section. All differences have the same capabilities as the original and are not expected to alter the experimental design.

## Provisions for quality control

All of the raw data, will be uploaded to the project page on the OSF (https://osf.io/xbign) and made publically available.

## Protocol 7: Western blot for phosphorylation of IGF-1R and expression of IGFBP3 in PC9-derived DTPs

This protocol utilizes Western blotting to compare the levels of phosphorylated IGF-1R and the expression of IGFBP3 in PC9 cells and PC9-derived DTPs. Non-phosphorylated IGF-1R is included as a loading control. It is a replication of Figure 7A.

### Sampling

This experiment will be repeated a total of 3 times.

The original data is qualitative, thus to determine an appropriate number of replicates to initially perform, sample sizes based on a range of potential variance was determined.

- See Power Calculations section for details.

Experiment has 2 cohorts:

- Cohort 1: Drug-naïve PC9 cells
- Cohort 2: Erlotinib-resistant PC9 DTPs

Western blotting is performed for the following proteins:

- Phosphorylated IGF-1R (Y1165,1166)
- IGF-1R
- IGFBP3

### Materials and reagents

| Reagent | Type | Manufacturer | Catalog # | Comments |
|---|---|---|---|---|
| PC9 Cells | Human cell line | Sigma-Aldrich | 90071810 | Originally obtained from Dr. Kazuto Nishio (National Cancer Center Hospital, Tokyo) |
| PC9 DTP cells | Human cell line | n/a | n/a | Generated according to protocol 2 |
| 10 cm dishes | Cell culture | Corning | 430167 | Original brand not specified. |
| RPMI 1640 (4.5 g/l glucose) | Cell culture | ATCC | 30-2001 | Original brand not specified. |
| Fetal bovine serum (FBS) | Cell culture | HyClone | SH30910.03 | Original brand not specified. |
| Penicillin-Streptomycin (5,000 U/ml) | Cell culture | Life Technologies | 15070-063 | Not originally reported. |
| Phosphate buffered saline (PBS) | Buffer | Life Technologies | 14190 | Original brand not specified. |
| RIPA buffer | Buffer | Sigma-Aldrich | R0278 | From replicating lab protocol |
| Complete mini protease inhibitor cocktail tablets | Inhibitor | Roche Diagnostics | 11 836 153 001 | From replicating lab protocol |
| Halt phosphatase inhibitor cocktail | Inhibitor | Thermo Scientific | 1862495 | From replicating lab protocol |
| Phenylmethanesulfonyl fluoride solution | Inhibitor | Sigma-Aldrich | 93482 | |
| NuPAGE LDS sample buffer (4X) | Western blot reagent | Life Technologies | NP0007 | Replaces Laemmli sample buffer |
| NuPage Sample Reducing Agent (10X) | Western blot reagent | Life Technologies | NP0004 | From replicating lab protocol |
| Molecular weight markers | Western blot reagent | Li-Cor | 928-40000 | Not originally reported. |
| NuPAGE 4-12% Bis-Tris gels (10 well/15 well) | Western blot reagent | Life Technologies | NP0335BOX/ NP0336BOX | From replicating lab protocol |
| NuPAGE MOPS SDS Running Buffer (20X) | Western blot reagent | Life Technologies | NP0001 | From replicating lab protocol |
| iBlot gel transfer stacks nitrocellulose | Western blot reagent | Life Technologies | IB301002 | From replicating lab protocol |

*Continued on next page*

*Continued*

| Reagent | Type | Manufacturer | Catalog # | Comments |
|---------|------|--------------|-----------|----------|
| Rabbit anti-human pIGF-1R (Y1165/1166) antibody | Antibodies | Abcam | ab192214 | Original brand not specified. |
| Mouse anti-IGF-1R antibody (clone JBW902) | Antibodies | EMD Millipore | 05-656 | Replaces GenScript (discontinued) |
| Rabbit anti-IGFBP3 antibody | Antibodies | Cell Signaling Technology | 13216 | Replaces GenScript (discontinued) |
| Donkey anti-mouse IRDye 680RD | Antibodies | Li-Cor | 926-68072 | Replaced HRP conjugated antibodies |
| Donkey anti-rabbit IRDye 800CW | Antibodies | Li-Cor | 926-32213 | |
| Odyssey Infrared Imager | Instrument | Li-Cor Biosciences | CLx | |
| Image Studio | Software | Li-Cor Biosciences | | |

## Procedure
### Note

- PC9 cells are grown in RPMI supplemented with 4.5 g/l glucose, 5% FBS and 1% penicillin/streptomycin at 37°C in a humidified atmosphere at 5% $CO_2$.
- Cells will be sent for mycoplasma testing and STR profiling.

1. Generate PC9-derived DTPs with 2 µM erlotinib in 10 cm tissue culture dishes, as described in protocol 2.
2. Plate drug naïve PC9 cells in 10 cm tissue culture dishes 2 days prior to harvest at sparse density.
   a. Seed cells at a density that will allow sufficient cells for analysis while remaining sub-confluent.
3. Dissociate cells from plates and count. Harvest drug naïve PC9 and PC9-derived DTPs in complete lysis buffer following replicating labs standard procedure.
   a. The detachment method identified in protocol 1, step 4 will be used.
   b. Complete lysis buffer: RIPA lysis buffer supplemented with 1X phosphatase inhibitor, protease inhibitor cocktail, 1 mM PMSF.
4. Normalize gel loading to total cell number, add 4X LDS sample buffer supplemented with reducing agent, and denature at 70°C for 10 min.
   a. Determine protein concentration by BCA assay following manufacturer's instructions for lysates from drug-naïve PC9 cells to determine the total protein concentrations relative to total cell number. Since erlotinib-resistant PC9 DTPs are anticipated to be less abundant, the PC9 BCA results will be used to approximate the concentration of DTPs to ensure enough protein is loaded.
5. Separate equivalent number of cells (~10–60 µg of protein) per lane with protein ladder and transfer to a membrane using the replicating labs standard procedures.
6. After transfer, block non-specific binding and immunoblot membrane with the following combinations of primary antibodies at the dilution/concentration recommended by the supplier.
   a. Rabbit anti-phospho-IGF-1R (Y1165/1166) (95 kDa) at a 1:500-1:2000 and mouse anti-IGF-1R (95 kDa) at 0.1–2 µg/ml.
   b. Rabbit anti-IGFBP3 (40 kDa) at a 1:1000 dilution and mouse anti-IGF-1R (95 kDa) at 0.1–2 µg/ml.

**Protocol 7 Western Blot Antibody Multiplexing**

| Combination | Protein of interest | | Loading Control | |
| | Description | Working Conc. | Description | Working Conc. |
|---|---|---|---|---|
| 1 | Rabbit anti-phospho-IGF-1R (Y1165/1166) (95 kDa) | 1:500-1:2000 | Mouse anti-IGF-1R (95 kDa) | 0.1–2 µg/ml |

| | 2 | Rabbit anti-IGFBP3 (40 kDa) | 1:1000 | Mouse anti-IGF-1R (95 kDa) | 0.1–2 µg/ml |
| --- | --- | --- | --- | --- | --- |

7. Wash and apply appropriate secondary antibodies for 1 hr at RT with constant agitation and detect signal using Odyssey imaging system.
8. Analyze bands with Image Studio software and normalize to loading controls.
   a. pIGF-1R (Y1165/1166) normalized to IGF-1R (total)
   b. IGFBP3 normalized to IGF-1R (total)
9. Repeat independently two additional times.

## Deliverables

- Data to be collected:
  - Full scans of each Western blot with ladder. (Compare to Figure 7A)
  - Raw data of band analysis and normalized bands for each sample.

## Confirmatory analysis plan

- Statistical Analysis of the Replication Data:
  - One-way MANOVA of normalized pIGF-1R and IGFBP3 levels of drug-naïve PC9 and erlotinib-resistant PC9 DTPs cells with the following planned comparisons using the Bonferroni correction:
    - Normalized pIGF-1R (Y1165/1166) levels from drug-naïve PC9 cells compared to erlotinib-resistant PC9 DTPs.
    - Normalized IGFBP3 levels from drug-naïve PC9 cells compared to erlotinib-resistant PC9 DTPs.
- Meta-analysis of original and replication attempt effect sizes:
  - The replication data (mean and 95% confidence interval) will be plotted with the original reported data value plotted as a single point on the same plot for comparison.

## Known differences from the original study

This replication attempt will prepare cells in RIPA lysis buffer while it is unclear if the original study used a lysis buffer or lysed cells directly in Laemmli sample buffer. All known differences are listed in the materials and reagents section above with the originally used item listed in the comments section. All differences have the same capabilities as the original and are not expected to alter the experimental design.

## Provisions for quality control

The samples will be normalized based on total cell number, similar to the original report, but protein concentration will also be determined to ensure enough lysate is loaded to ensure detection of the proteins of interest. All of the raw data will be uploaded to the project page on the OSF (https://osf.io/xbign) and made publically available.

## Protocol 8: Western blot for phosphorylation of IGF-1R following IGF-1R inhibition

This protocol utilizes Western blotting to compare the levels of phosphorylated IGF-1R upon treatment of PC9-derived DTPs with the IGF-1R inhibitor AEW541. ERK1/2 is used as a loading control. It is a replication of Figure 7C.

### Sampling

This experiment will be repeated a total of 3 times.

The original data is qualitative, thus to determine an appropriate number of replicates to initially perform, sample sizes based on a range of potential variance was determined.

- See Power Calculations section for details.

Experiment has 2 cohorts:

- Cohort 1: Erlotinib-resistant PC9 DTPs + vehicle
- Cohort 2: Erlotinib-resistant PC9 DTPs + 1 µM AEW541 for 2 hr

Western blotting is performed for the following proteins:

- Phospho-IGF-1R (Y1165/1166)
- IGF-1R [additional control]
- ERK1/2

## Materials and reagents

| Reagent | Type | Manufacturer | Catalog # | Comments |
|---|---|---|---|---|
| PC9 DTP cells | Human cell line | n/a | n/a | Generated according to protocol 2 |
| RPMI 1640 (4.5 g/l glucose) | Cell culture | ATCC | 30-2001 | Original brand not specified. |
| Fetal bovine serum (FBS) | Cell culture | HyClone | SH30910.03 | Original brand not specified. |
| Penicillin-Streptomycin (5,000 U/ml) | Cell culture | Life Technologies | 15070-063 | Not originally reported. |
| Phosphate buffered saline (PBS) | Buffer | Life Technologies | 14190 | Original brand not specified. |
| AEW541 | Inhibitor | Cayman Chemical | 13641 | Replaces original obtained from Novartis Pharmaceuticals |
| DMSO | Chemical | Sigma-Aldrich | D8418 | Not originally reported. |
| RIPA buffer | Buffer | Sigma-Aldrich | R0278 | From replicating lab protocol |
| Complete mini protease inhibitor cocktail tablets | Inhibitor | Roche Diagnostics | 11 836 153 001 | From replicating lab protocol |
| Halt phosphatase inhibitor cocktail | Inhibitor | Thermo Scientific | 1862495 | From replicating lab protocol |
| Phenylmethanesulfonyl fluoride solution | Inhibitor | Sigma-Aldrich | 93482 | |
| NuPAGE LDS sample buffer (4X) | Western blot reagent | Life Technologies | NP0007 | Replaces Laemmli sample buffer |
| NuPage Sample Reducing Agent (10X) | Western blot reagent | Life Technologies | NP0004 | From replicating lab protocol |
| Molecular weight markers | Western blot reagent | Li-Cor | 928-40000 | Not originally reported. |
| NuPAGE 4-12% Bis-Tris gels (10 well/15 well) | Western blot reagent | Life Technologies | NP0335BOX/ NP0336BOX | From replicating lab protocol |
| NuPAGE MOPS SDS Running Buffer (20X) | Western blot reagent | Life Technologies | NP0001 | From replicating lab protocol |
| iBlot gel transfer stacks nitrocellulose | Western blot reagent | Life Technologies | IB301002 | From replicating lab protocol |
| Rabbit anti-human pIGF-1R (Y1165/1166) antibody | Antibodies | Abcam | ab192214 | Original brand not specified. Recommended working dilution: 1:500-1:2000 |
| Mouse anti-IGF-1R antibody (clone JBW902) | Antibodies | EMD Millipore | 05-656 | Replaces GenScript (discontinued) Recommended working concentration: 0.1-2 µg/ml |
| Mouse anti-ERK1/2 antibody (clone L34F12) | Antibodies | Cell Signaling Technology | 4696 | Original catalog number not specified. Recommended working dilution: 1:2000 |
| Donkey anti-mouse IRDye 680RD | Antibodies | Li-Cor | 926-68072 | Replaced HRP conjugated antibodies |
| Donkey anti-rabbit IRDye 800CW | Antibodies | Li-Cor | 926-32213 | |
| Odyssey Infrared Imager | Instrument | Li-Cor Biosciences | CLx | |
| Image Studio | Software | Li-Cor Biosciences | | |

## Procedure

### Note

- PC9 cells are grown in RPMI supplemented with 4.5 g/l glucose, 5% FBS and 1% penicillin/streptomycin at 37°C in a humidified atmosphere at 5% $CO_2$.

1. Generate PC9-derived DTPs with 2 µM erlotinib in 10 cm tissue culture dishes, as described in protocol 2.
2. Treat cells with DMSO (vehicle) or 1 µM AEW541 for 2 hr.
   a. Stock concentration used will be determined from assessing DMSO solvent toxicity on PC9 cells (protocol 1, step 3) and drug-withdrawn DTPs (protocol 1, step 6) to determine final DMSO concentration.
3. Dissociate cells from plates and count. Harvest drug naïve PC9 and PC9-derived DTPs in complete lysis buffer following replicating labs standard procedure.
   a. The detachment method identified in protocol 1, step 4 will be used.
   b. Complete lysis buffer: RIPA lysis buffer supplemented with 1X phosphatase inhibitor, protease inhibitor cocktail, 1 mM PMSF.
4. Normalize gel loading to total cell number, add 4X LDS sample buffer supplemented with reducing agent, and denature at 70°C for 10 min.
5. Separate equivalent number of cells (~10–60 µg of protein) per lane with protein ladder and transfer to a membrane using the replicating labs standard procedures.
6. After transfer, block non-specific binding and immunoblot membrane with the following combinations of primary antibodies at the dilution/concentration recommended by the supplier.
   a. Rabbit anti-phospho-IGF-1R (Y1165/1166) (95 kDa) at a 1:500-1:2000 dilution and mouse anti-ERK1/2 (42/44 kDa) at a 1:2000 dilution.
   b. Rabbit anti-phospho-IGF-1R (Y1165/1166) (95 kDa) at a 1:500-1:2000 dilution and mouse anti-IGF-1R at 0.1-2 µg/ml.

**Protocol 8 Western Blot Antibody Multiplexing**

| | Protein of interest | | Loading Control | |
|---|---|---|---|---|
| Combination | Description | Working Conc. | Description | Working Conc. |
| 1 | Rabbit phospho-IGF-1R (Y1165/1166) (95 kDa) | 1:500-1:2000 | Mouse anti-ERK1/2 (42/44 kDa) | 1:2000 |
| 2 | Rabbit phospho-IGF-1R (Y1165/1166) (95 kDa) | 1:500-1:2000 | Mouse anti-IGF-1R (95 kDa) | 0.1–2 µg/ml |

7. Wash and apply appropriate secondary antibodies for 1 hr at RT with constant agitation and detect signal using Odyssey imaging system.
8. Analyze bands with Image Studio software and normalize to loading controls.
   a. pIGF-1R (Y1165/1166) normalized to ERK1/2 (total)
   b. pIGF-1R (Y1165/1166) normalized to IGF-1R (total) [additional]
9. Repeat independently two additional times.

### Deliverables

- Data to be collected:
  - Full scans of each Western blot with ladder. (Compare to Figure 7C)
  - Raw data of band analysis and normalized bands for each sample.

### Confirmatory analysis plan

- Statistical Analysis of the Replication Data:
  - Unpaired two-tailed *t*-test of pIGF-1R (Y1165/1166) levels normalized to ERK1/2 from erlotinib-resistant PC9 DTPs treated with vehicle compared to AEW541.
- Meta-analysis of original and replication attempt effect sizes:
  - The replication data (mean and 95% confidence interval) will be plotted with the original reported data value plotted as a single point on the same plot for comparison.
- Additional exploratory analysis:

- pIGF-1R levels will also be normalized to IGF-1R (total) and the same analysis described above will be performed, which serves as an independent normalization control not included in the original report.

## Known differences from the original study

An additional control, total IGF-1R, was added to this replication attempt that was not originally reported in Figure 7C to normalize the phosphorylated IGF-1R levels. The original loading control of ERK1/2 will also be utilized to allow for a direct comparison. This replication attempt will pre-pare cells in RIPA lysis buffer, while it is unclear if the original study used a lysis buffer or lysed cells directly in Laemmli sample buffer. All known differences are listed in the materials and reagents section above with the originally used item listed in the comments section. All differences have the same capabilities as the original and are not expected to alter the experimental design.

## Provisions for quality control

All of the raw data will be uploaded to the project page on the OSF (https://osf.io/xbign) and made publically available.

## Protocol 9: Western blot for dimethylated H3K4 following IGF-1R inhibition

This protocol utilizes Western blotting to compare the levels of the histone demethylase KDM5A in PC9-derived DTPs in the presence or absence of the IGF-1R inhibitor AEW541. ERK1/2 protein is included as a loading control. It is a replication of Figure 7I.

## Sampling

This experiment will be repeated a total of 3 times.

The original data is qualitative, thus to determine an appropriate number of replicates to initially perform, sample sizes based on a range of potential variance was determined.

- See Power Calculations section for details.

Experiment has 2 conditions:

- Cohort 1: Erlotinib-resistant PC9 DTPs + vehicle
- Cohort 2: Erlotinib-resistant PC9 DTPs + 1 µM AEW541 for 24 hr

Western blotting is performed for the following proteins:

- KDM5A
- ERK1/2

## Materials and reagents

| Reagent | Type | Manufacturer | Catalog # | Comments |
|---------|------|--------------|-----------|----------|
| PC9 DTP cells | Human cell line | n/a | n/a | Generated according to protocol 2 |
| RPMI 1640 (4.5 g/l glucose) | Cell culture | ATCC | 30-2001 | Original brand not specified. |
| Fetal bovine serum (FBS) | Cell culture | HyClone | SH30910.03 | Original brand not specified. |
| Penicillin-Streptomycin (5,000 U/ml) | Cell culture | Life Technologies | 15070-063 | Not originally reported. |
| Phosphate buffered saline (PBS) | Buffer | Life Technologies | 14190 | Original brand not specified. |
| AEW541 | Inhibitor | Cayman Chemical | 13641 | Replaces original obtained from Novartis Pharmaceuticals |
| DMSO | Chemical | Sigma-Aldrich | D8418 | Not originally reported. |

*Continued on next page*

*Continued*

| Reagent | Type | Manufacturer | Catalog # | Comments |
|---|---|---|---|---|
| RIPA buffer | Buffer | Sigma-Aldrich | R0278 | From replicating lab protocol |
| Complete mini protease inhibitor cocktail tablets | Inhibitor | Roche Diagnostics | 11 836 153 001 | From replicating lab protocol |
| Halt phosphatase inhibitor cocktail | Inhibitor | Thermo Scientific | 1862495 | From replicating lab protocol |
| Phenylmethanesulfonyl fluoride solution | Inhibitor | Sigma-Aldrich | 93482 | |
| NuPAGE LDS sample buffer (4X) | Western blot reagent | Life Technologies | NP0007 | Replaces Laemmli sample buffer |
| NuPage Sample Reducing Agent (10X) | Western blot reagent | Life Technologies | NP0004 | From replicating lab protocol |
| Molecular weight markers | Western blot reagent | Li-Cor | 928-40000 | Not originally reported. |
| NuPAGE 4-12% Bis-Tris gels (10 well/15 well) | Western blot reagent | Life Technologies | NP0335BOX/ NP0336BOX | From replicating lab protocol |
| NuPAGE MOPS SDS Running Buffer (20X) | Western blot reagent | Life Technologies | NP0001 | From replicating lab protocol |
| iBlot gel transfer stacks nitrocellulose | Western blot reagent | Life Technologies | IB301002 | From replicating lab protocol |
| Rabbit anti-KDM5A antibody | Antibodies | Bethyl Laboratories | A300-897A | Original catalog number not specified. |
| Mouse anti-ERK1/2 antibody (clone L34F12) | Antibodies | Cell Signaling Technology | 4696 | Original catalog number not specified. Recommended working dilution: 1:2000 |
| Donkey anti-mouse IRDye 680RD | Antibodies | Li-Cor | 926-68072 | Replaced HRP conjugated antibodies |
| Donkey anti-rabbit IRDye 800CW | Antibodies | Li-Cor | 926-32213 | |
| Odyssey Infrared Imager | Instrument | Li-Cor Biosciences | CLx | |
| Image Studio | Software | Li-Cor Biosciences | | |

## Procedure

### Note

- PC9 cells are grown in RPMI supplemented with 4.5 g/l glucose, 5% FBS and 1% penicillin/ streptomycin at 37°C in a humidified atmosphere at 5% $CO_2$.

1. Generate PC9-derived DTPs with 2 µM erlotinib in 10 cm tissue culture dishes, as described in protocol 2.
2. Treat cells with DMSO (vehicle) or 1 µM AEW541 for 24 hr.
   a. Stock concentration used will be determined from assessing DMSO solvent toxicity on PC9 cells (protocol 1, step 3) and drug-withdrawn DTPs (protocol 1, step 6) to determine final DMSO concentration.
3. Dissociate cells from plates and count. Harvest drug naïve PC9 and PC9-derived DTPs in complete lysis buffer following replicating labs standard procedure.
   a. The detachment method identified in protocol 1, step 4 will be used.
   b. Complete lysis buffer: RIPA lysis buffer supplemented with 1X phosphatase inhibitor, protease inhibitor cocktail, 1 mM PMSF.
4. Normalize gel loading to total cell number, add 4X LDS sample buffer supplemented with reducing agent, and denature at 70°C for 10 min.
5. Separate equivalent number of cells (~10–60 µg of protein) per lane with protein ladder and transfer to a membrane using the replicating labs standard procedures.

6. After transfer, block non-specific binding and immunoblot membrane with the following combinations of primary antibodies at the dilution/concentration recommended by the supplier.
    a. Rabbit anti-KDM5A (200 kDa) at a 1:2000 to 1:10,000 dilution and mouse anti-ERK1/2 (42/44 kDa) at a 1:2000 dilution.

**Protocol 9 Western Blot Antibody Multiplexing**

| | Protein of interest | | Loading Control | |
|---|---|---|---|---|
| Combination | Description | Working Conc. | Description | Working Conc. |
| 1 | Rabbit anti-KDM5A (200 kDa) | 1:2000 to 1:10,000 | Mouse anti-ERK1/2 (42/44 kDa) | 1:1000 |

7. Wash and apply appropriate secondary antibodies for 1 hr at RT with constant agitation and detect signal using Odyssey imaging system.
8. Analyze bands with Image Studio software and normalize to loading controls.
    a. KDM5A normalized to ERK1/2 (total)
9. Repeat independently two additional times.

## Deliverables

- Data to be collected:
    - Full scans of each Western blot with ladder. (Compare to Figure 7I)
    - Raw data of band analysis and normalized bands for each sample.

## Confirmatory analysis plan

- Statistical Analysis of the Replication Data:
    - Unpaired two-tailed *t*-test of normalized KDM5A levels from erlotinib-resistant PC9 DTPs treated with vehicle compared to AEW541.
- Meta-analysis of original and replication attempt effect sizes:
    - The replication data (mean and 95% confidence interval) will be plotted with the original reported data value plotted as a single point on the same plot for comparison.

## Known differences from the original study

This replication attempt will prepare cells in RIPA lysis buffer, while it is unclear if the original study used a lysis buffer or lysed cells directly in Laemmli sample buffer. All known differences are listed in the materials and reagents section above with the originally used item listed in the comments section. All differences have the same capabilities as the original and are not expected to alter the experimental design.

## Provisions for quality control

All of the raw data will be uploaded to the project page on the OSF (https://osf.io/xbign) and made publically available.

## Power Calculations

For additional details on power calculations, please see analysis scripts and associated files on the Open Science Framework:
    https://osf.io/q9bxy/

## Protocol 1

Not applicable

## Protocol 2

Not applicable

## Protocol 3

The original data (Figure 2E) reported no difference between the absolute $IC_{50}$ values (50% cell killing). Thus this is a sensitivity calculation to determine the detectable effect size with the planned sample size.

### Test family

- Two-tailed $t$ test, difference between two independent means, alpha error = 0.05

### Power Calculations

- Performed with G*Power software, version 3.1.7 (*Faul et al., 2007*).

| Group 1 | Group 2 | Detectable effect size $d$ (based on sample size) | A priori power | Group 1 sample size | Group 2 sample size |
|---|---|---|---|---|---|
| Drug-naïve PC9 cells | PC9 DTPs cultured in drug-free medium | 2.38075 | 80.0% | 4[1] | 4[1] |

[1] The sample size is the same as originally reported.

## Protocol 4

The original data presented is qualitative (images of Western blots). We used Image Studio Lite v. 4.0.21 (LI-COR) to perform densitometric analysis of the presented bands to quantify the original effect size where possible. The data presented in Figures 1E, upper right panel, and Figure 2B were unable to be quantified for all bands and are thus considered exploratory in nature. Due to the lack of raw original data, and inability to quantify some of the Western blots, we are unable to perform power calculations for all comparisons. However, because the same samples will be used to detect each protein of interest (3 total), the alpha error will be adjusted to account for multiple comparisons.

Summary of original data quantified from the image reported in Figure 4A, upper panel:

| Cells | Normalized H3K14Ac signal (normalized to GAPDH) |
|---|---|
| Drug-naïve PC9 cells | 0.7245 |
| PC9 DTPs | 0.1674 |

The original data does not indicate the error associated with multiple biological replicates. To identify a suitable sample size, power calculations were performed using different levels of relative variance using the values quantified from the reported image as the mean. At each level of variance the effect size was estimated and used to calculate the needed sample size to achieve at least 80% power with the indicated alpha error. The achieved power is reported.

H3K14Ac normalized signal:

### Test family

- Two-tailed $t$ test, difference between two independent means, Bonferroni's correction, alpha error = 0.01667

### Power calculations

- Performed with G*Power software, version 3.1.7 (*Faul et al., 2007*).

**2% variance:**

| Group 1 | Group 2 | Effect size $d$ | A priori power | Group 1 sample size | Group 2 sample size |
|---|---|---|---|---|---|
| Drug-naïve PC9 cells | PC9 DTPs | 52.97365 | 99.9% | 2 | 2 |

**15% variance:**

| Group 1 | Group 2 | Effect size $d$ | A priori power | Group 1 sample size | Group 2 sample size |
|---|---|---|---|---|---|
| Drug-naïve PC9 cells | PC9 DTPs | 7.06315 | 99.7% | 3 | 3 |

**28% variance:**

| Group 1 | Group 2 | Effect size $d$ | A priori power | Group 1 sample size | Group 2 sample size |
|---|---|---|---|---|---|
| Drug-naïve PC9 cells | PC9 DTPs | 3.78383 | 94.4% | 4 | 4 |

**40% variance:**

| Group 1 | Group 2 | Effect size $d$ | A priori power | Group 1 sample size | Group 2 sample size |
|---|---|---|---|---|---|
| Drug-naïve PC9 cells | PC9 DTPs | 2.64868 | 84.5% | 5 | 5 |

CD133 and pEGFR (Y1068) normalized signals:

## Test family

- Two-tailed *t* test, difference between two independent means, Bonferroni's correction, alpha error = 0.01667

## Power calculations

- Performed with G*Power software, version 3.1.7 (*Faul et al., 2007*).

| Group 1 | Group 2 | Detectable effect size $d$ (based on sample size) | A priori power | Group 1 sample size | Group 2 sample size |
|---|---|---|---|---|---|
| Drug-naïve PC9 cells | PC9 DTPs | 3.03763[1] | 80.0%[1] | 4[2] | 4[2] |

[1] A sensitivity calculation was performed since the original data presented in Figures 1E, upper right panel, and Figure 2B were unable to be quantified for all bands. This is the effect size that can be detected with 80% power and the indicated sample size.

[2] The sample size is the number of replicates initially performed based on the H3K14Ac analysis.

In order to produce quantitative replication data, we will run the experiment four times. Each time we will quantify band intensity. We will determine the standard deviation of band intensity across the biological replicates and combine this with the reported value from the original study to simulate the original effect size. We will use this simulated effect size to determine the number of replicates necessary to reach a power of at least 80%. We will then perform additional replicates, if required, to ensure that the experiment has more than 80% power to detect the original effect.

## Protocol 5
## Cell cycle analysis

Summary of original data estimated from graph reported in Figure 1B, lower panel:

| Cells | Cell cycle phase | Mean % of cells | Stdev | N |
|---|---|---|---|---|
| Drug-naïve PC9 cells | G1 | 39 | 5 | 2 |
| | S | 45 | 3 | 2 |
| | G2/M | 12 | 2 | 2 |
| PC9 DTPs | G1 | 73 | 4 | 2 |
| | S | 7 | 4 | 2 |
| | G2/M | 3 | 1 | 2 |

## Test family

- Proportions: Cochran-Mantel-Haenszel test: alpha error = 0.05.

## Power calculations

- Performed with R software, version 3.1.2 (*Team, 2014*).

| Group 1 | Group 2 | Number of simulations | A priori power | Group 1 sample size | Group 2 sample size |
|---|---|---|---|---|---|
| Drug-naïve PC9 cells | PC9 DTPs | 10,000[1] | 99.9% | 2[2] | 2[2] |

[1] The estimated data was used to create simulated data sets with preserved sampling structure assuming a normal distribution. For a given $n$ (the number of replicates) 10,000 simulations were run and Mantel-Haenszel chi-squared value was calculated for each simulated data set. The power was then calculated by counting the number of times $p \leq 0.05$ and dividing by 10,000.
[2] A sample size of 3 per group will be used based on the percent of sub-G1 cells analysis.

## Sub-G1 analysis

Summary of original data estimated from graph reported in Figure 4B, upper panel:

| Cells | TSA concentration | Mean % of sub-G1 cells | Stdev | N |
|---|---|---|---|---|
| Drug-naïve PC9 cells | 0 nM | 1.5 | 0.2 | 3 |
| | 50 nM | 2 | 0.8 | 3 |
| | 100 nM | 2.6 | 0.6 | 3 |
| PC9 DTPs | 0 nM | 5.4 | 0.2 | 3 |
| | 50 nM | 17.5 | 1.5 | 3 |
| | 100 nM | 23.5 | 2 | 3 |

## Test family

- ANOVA: Fixed effects, special, main effects and interactions: alpha error = 0.05.

## Power calculations

- Performed with G*Power software, version 3.1.7 (*Faul et al., 2007*).
- ANOVA F test statistic and partial $\eta^2$ performed with R software, version 3.1.2 (*Team, 2014*).

| Groups | F test statistic | Partial $\eta^2$ | Effect size $f$ | A priori power | Total sample size |
|---|---|---|---|---|---|
| Drug-naïve PC9 cells and PC9 DTPs left untreated or treated with TSA (50 nM or 100 nM) | F(2,12) = 9.3683 (interaction) | 0.60959 | 1.24956 | 84.7%[1] | 12[1] (6 groups) |

[1] 18 total samples (3 per group) will be used based on the planned comparisons making the power 99.0%.

## Test family

- Two-tailed *t* test, difference between two independent means, Bonferorni's correction: alpha error = 0.0125

## Power calculations

- Performed with G*Power software, version 3.1.7 (*Faul et al., 2007*).

| Group 1 | Group 2 | Effect size $d$ | A priori power | Group 1 sample size | Group 2 sample size |
|---|---|---|---|---|---|
| Untreated drug-naïve PC9 cells | 50 nM TSA treated drug-naïve PC9 cells | 0.85749[1] | 80.0%[1] | 3 | 3 |
| Untreated drug-naïve PC9 cells | 100 nM TSA treated drug-naïve PC9 cells | 2.45967[1] | 80.0%[1] | 3 | 3 |
| Untreated PC9 DTPs | 50 nM TSA treated PC9 DTPs | 11.30792 | 99.9% | 3 | 3 |
| Untreated PC9 DTPs | 100 nM TSA treated PC9 DTPs | 12.73512 | 86.8%[2] | 2[2] | 2[2] |

[1] A sensitivity calculation was performed since the original data showed a non-significant effect. This is the effect size that can be detected with 80% power and the indicated sample size.

[2] 3 samples per group will be used based on the vehicle vs 50 nM TSA treated PC9 DTP comparison making the power 99.9%.

## Protocol 6

Summary of original data estimated from graph reported in Figure 5B:

| Dataset being analyzed | Mean | SD | N |
|---|---|---|---|
| Erlotinib | 525 | 8 | 3 |
| Erlotinib + TSA | 0 | 0 | 3 |
| Erlotinib + AEW541 | 10 | 1 | 3 |

## Test family

- Two-tailed *t* test: Means: Wilcoxon-Mann-Whitney, Bonferorni's correction: alpha error = 0.025

## Power calculations

- Performed with G*Power software, version 3.1.7. (*Faul et al., 2007*)

| Group 1 | Group 2 | Effect size *d* | A priori power | Group 1 sample size | Group 2 sample size |
|---------|---------|-----------------|----------------|---------------------|---------------------|
| Erlotinib | Erlotinib + TSA | 92.80777 | 99.9% | 2[1] | 2[1] |
| Erlotinib | Erlotinib + AEW541 | 90.33698 | 99.9% | 2[1] | 2[1] |

[1] A sample size of 3 per group will be used as a minimum sample size.

## Protocol 7

The original data presented is qualitative (images of Western blots). We used Image Studio Lite v. 4.0.21 (LI-COR) to perform densitometric analysis of the presented bands to quantify the original effect size where possible.

Summary of original data quantified from the image reported in Figure 7A:

| Cells | Normalized pIFG-1R signal (normalized to IGF-1R (total)) | Normalized IGFBP3 signal (normalized to IGF-1R (total)) |
|-------|-----------------------------------------------------------|----------------------------------------------------------|
| Drug-naïve PC9 cells | 0.1065 | 0.4276 |
| PC9 DTPs | 2.3508 | 2.5992 |

The original data does not indicate the error associated with multiple biological replicates. To identify a suitable sample size, power calculations were performed using different levels of relative variance using the values quantified from the reported image as the mean. At each level of variance the effect size was estimated and used to calculate the needed sample size to achieve at least 80% power with the indicated alpha error. The achieved power is reported.

### Test family

- Due to the lack of raw original data we are unable to perform power calculations using a MANOVA. We are determining sample size using a two-way ANOVA.
- ANOVA, Fixed effects, special, main effects and interactions, alpha error = 0.05

### Power calculations

- Performed with G*Power software, version 3.1.7 (*Faul et al., 2007*).
- ANOVA F test statistic and partial $\eta^2$ performed with R software, version 3.1.2 (*Team, 2014*).

**2% variance:**

| Groups | F test statistic | Partial $\eta^2$ | Effect size *f* | A priori power | Total sample size |
|--------|------------------|------------------|-----------------|----------------|-------------------|
| Drug-naïve PC9 cells and PC9 DTPs | F(1,8) = 11722 (main effect, cell type) | 0.99932 | 38.2789 | 99.9% | 8 (4 groups) |

**15% variance:**

| Groups | F test statistic | Partial $\eta^2$ | Effect size *f* | A priori power | Total sample size |
|--------|------------------|------------------|-----------------|----------------|-------------------|
| Drug-naïve PC9 cells and PC9 DTPs | F(1,8) = 208.39 (main effect, cell type) | 0.96303 | 5.10382 | 99.9% | 12 (4 groups) |

**28% variance:**

| Groups | F test statistic | Partial $\eta^2$ | Effect size $f$ | A priori power | Total sample size |
|---|---|---|---|---|---|
| Drug-naïve PC9 cells and PC9 DTPs | F(1,8) = 59.8066 (main effect, cell type) | 0.88202 | 2.73419 | 99.9% | 12 (4 groups) |

**40% variance:**

| Groups | F test statistic | Partial $\eta^2$ | Effect size $f$ | A priori power | Total sample size |
|---|---|---|---|---|---|
| Drug-naïve PC9 cells and PC9 DTPs | F(1,8) = 29.3052 (main effect, cell type) | 0.78555 | 1.91394 | 99.9% | 16 (4 groups) |

## pIFG-1R normalized signal
Test family

- Two-tailed $t$ test, difference between two independent means, Bonferroni's correction, alpha error = 0.025

Power calculations

- Performed with G*Power software, version 3.1.7 (*Faul et al., 2007*).

**2% variance:**

| Group 1 | Group 2 | Effect size $d$ | A priori power | Group 1 sample size | Group 2 sample size |
|---|---|---|---|---|---|
| Drug-naïve PC9 cells | PC9 DTPs | 67.43731 | 99.9% | 2 | 2 |

**15% variance:**

| Group 1 | Group 2 | Effect size $d$ | A priori power | Group 1 sample size | Group 2 sample size |
|---|---|---|---|---|---|
| Drug-naïve PC9 cells | PC9 DTPs | 8.99164 | 86.8% | 2 | 2 |

**28% variance:**

| Group 1 | Group 2 | Effect size $d$ | A priori power | Group 1 sample size | Group 2 sample size |
|---|---|---|---|---|---|
| Drug-naïve PC9 cells | PC9 DTPs | 4.81695 | 94.8% | 3 | 3 |

**40% variance:**

| Group 1 | Group 2 | Effect size $d$ | A priori power | Group 1 sample size | Group 2 sample size |
|---|---|---|---|---|---|
| Drug-naïve PC9 cells | PC9 DTPs | 3.37187 | 92.7% | 4 | 4 |

## IGFBP3 normalized signal
### Test family

- Two-tailed *t* test, difference between two independent means, Bonferroni's correction, alpha error = 0.025

### Power calculations

- Performed with G*Power software, version 3.1.7 (*Faul et al., 2007*).

**2% variance:**

| Group 1 | Group 2 | Effect size *d* | A priori power | Group 1 sample size | Group 2 sample size |
|---|---|---|---|---|---|
| Drug-naïve PC9 cells | PC9 DTPs | 58.29364 | 99.9% | 2 | 2 |

**15% variance:**

| Group 1 | Group 2 | Effect size *d* | A priori power | Group 1 sample size | Group 2 sample size |
|---|---|---|---|---|---|
| Drug-naïve PC9 cells | PC9 DTPs | 7.77249 | 99.9% | 3 | 3 |

**28% variance:**

| Group 1 | Group 2 | Effect size *d* | A priori power | Group 1 sample size | Group 2 sample size |
|---|---|---|---|---|---|
| Drug-naïve PC9 cells | PC9 DTPs | 4.16383 | 87.6% | 3 | 3 |

| Group 1 | Group 2 | Effect size *d* | A priori power | Group 1 sample size | Group 2 sample size |
|---|---|---|---|---|---|
| Drug-naïve PC9 cells | PC9 DTPs | 2.91468 | 83.6% | 4 | 4 |

In order to produce quantitative replication data, we will run the experiment three times. Each time we will quantify band intensity. We will determine the standard deviation of band intensity across the biological replicates and combine this with the reported value from the original study to simulate the original effect size. We will use this simulated effect size to determine the number of replicates necessary to reach a power of at least 80%. We will then perform additional replicates, if required, to ensure that the experiment has more than 80% power to detect the original effect.

### Protocol 8

The original data presented is qualitative (images of Western blots). We used Image Studio Lite v. 4.0.21 (LI-COR) to perform densitometric analysis of the presented bands to quantify the original effect size where possible.

Summary of original data quantified from the image reported in Figure 7C:

| Cells/Treatment | Normalized pIFG-1R signal (normalized to ERK1/2 (total)) |
|---|---|
| Vehicle treated PC9 DTPs | 1.0701 |
| AEW541 treated PC9 DTPs | 0.0906 |

The original data does not indicate the error associated with multiple biological replicates. To identify a suitable sample size, power calculations were performed using different levels of relative variance using the values quantified from the reported image as the mean. At each level of variance the effect size was estimated and used to calculate the needed sample size to achieve at least 80% power with the indicated alpha error. The achieved power is reported.

## pIFG-1R normalized signal
## Test family

- ▪ Two-tailed *t* test, difference between two independent means,, alpha error = 0.05

## Power calculations

- ▪ Performed with G*Power software, version 3.1.7 (*Faul et al., 2007*).

**2% variance:**

| Group 1 | Group 2 | Effect size *d* | A priori power | Group 1 sample size | Group 2 sample size |
|---|---|---|---|---|---|
| Vehicle treated PC9 DTPs | AEW541 treated PC9 DTPs | 64.49516 | 99.9% | 2 | 2 |

**15% variance:**

| Group 1 | Group 2 | Effect size *d* | A priori power | Group 1 sample size | Group 2 sample size |
|---|---|---|---|---|---|
| Vehicle treated PC9 DTPs | AEW541 treated PC9 DTPs | 8.59935 | 97.4% | 2 | 2 |

**28% variance:**

| Group 1 | Group 2 | Effect size *d* | A priori power | Group 1 sample size | Group 2 sample size |
|---|---|---|---|---|---|
| Vehicle treated PC9 DTPs | AEW541 treated PC9 DTPs | 4.60680 | 98.3% | 3 | 3 |

**40% variance:**

| Group 1 | Group 2 | Effect size *d* | A priori power | Group 1 sample size | Group 2 sample size |
|---|---|---|---|---|---|
| Vehicle treated PC9 DTPs | AEW541 treated PC9 DTPs | 3.22476 | 83.5% | 3 | 3 |

In order to produce quantitative replication data, we will run the experiment three times. Each time we will quantify band intensity. We will determine the standard deviation of band intensity across the biological replicates and combine this with the reported value from the original study to simulate the original effect size. We will use this simulated effect size to determine the number of replicates necessary to reach a power of at least 80%. We will then perform additional replicates, if required, to ensure that the experiment has more than 80% power to detect the original effect.

## Protocol 9

The original data presented is qualitative (images of Western blots). We used Image Studio Lite v. 4.0.21 (LI-COR) to perform densitometric analysis of the presented bands to quantify the original effect size where possible.

Summary of original data quantified from the image reported in Figure 7I:

| Cells/Treatment | Normalized KDM5A signal (normalized to ERK1/2 (total)) |
| --- | --- |
| Vehicle treated PC9 DTPs | 0.5432 |
| AEW541 treated PC9 DTPs | 0.0314 |

The original data does not indicate the error associated with multiple biological replicates. To identify a suitable sample size, power calculations were performed using different levels of relative variance using the values quantified from the reported image as the mean. At each level of variance the effect size was estimated and used to calculate the needed sample size to achieve at least 80% power with the indicated alpha error. The achieved power is reported.

## KDM5A normalized signal
### Test family

- Two-tailed $t$ test, difference between two independent means, alpha error = 0.05

### Power calculations

- Performed with G*Power software, version 3.1.7 (*Faul et al., 2007*).

**2% variance:**

| Group 1 | Group 2 | Effect size *d* | A priori power | Group 1 sample size | Group 2 sample size |
| --- | --- | --- | --- | --- | --- |
| Vehicle treated PC9 DTPs | AEW541 treated PC9 DTPs | 66.51393 | 99.9% | 2 | 2 |

**15% variance:**

| Group 1 | Group 2 | Effect size *d* | A priori power | Group 1 sample size | Group 2 sample size |
| --- | --- | --- | --- | --- | --- |
| Vehicle treated PC9 DTPs | AEW541 treated PC9 DTPs | 8.86852 | 97.9% | 2 | 2 |

**28% variance:**

| Group 1 | Group 2 | Effect size *d* | A priori power | Group 1 sample size | Group 2 sample size |
| --- | --- | --- | --- | --- | --- |
| Vehicle treated PC9 DTPs | AEW541 treated PC9 DTPs | 4.75099 | 98.8% | 3 | 3 |

**40% variance:**

| Group 1 | Group 2 | Effect size *d* | A priori power | Group 1 sample size | Group 2 sample size |
| --- | --- | --- | --- | --- | --- |
| Vehicle treated PC9 DTPs | AEW541 treated PC9 DTPs | 3.32570 | 85.5% | 3 | 3 |

In order to produce quantitative replication data, we will run the experiment three times. Each time we will quantify band intensity. We will determine the standard deviation of band intensity across the biological replicates and combine this with the reported value from the original study to simulate the original effect size. We will use this simulated effect size to determine the number of replicates necessary to reach a power of at least 80%. We will then perform additional replicates, if required, to ensure that the experiment has more than 80% power to detect the original effect.

## Acknowledgements

The Reproducibility Project: Cancer Biology core team would like to thank the original authors, in particular Jeffrey Settleman, for generously sharing critical information to ensure the fidelity and quality of this replication attempt. We thank Courtney Soderberg at the Center for Open Science for assistance with statistical analyses. We would also like to thank the following companies for generously donating reagents to the Reproducibility Project: Cancer Biology; American Type Culture Collection (ATCC), Applied Biological Materials, BioLegend, Charles River Laboratories, Corning Incorporated, DDC Medical, EMD Millipore, Harlan Laboratories, LI-COR Biosciences, Mirus Bio, Novus Biologicals, Sigma-Aldrich, and System Biosciences (SBI).

## Additional information

### Group author details

Reproducibility Project: Cancer Biology

Elizabeth Iorns: Science Exchange, Palo Alto, United States; William Gunn: Mendeley, London, United Kingdom; Fraser Tan: Science Exchange, Palo Alto, United States; Joelle Lomax: Science Exchange, Palo Alto, United States; Nicole Perfito: Science Exchange, Palo Alto, United States; Timothy Errington: Center for Open Science, Charlottesville, United States

### Competing interests

BH, EH, CD, MS: This is a Science Exchange associated laboratory. RP:CB: We disclose that EI, FT, JL, NP are employed by and hold shares in Science Exchange Inc. The other authors declare that no competing interests exist.

### Funding

| Funder | Author |
| --- | --- |
| Laura and John Arnold Foundation | Reproducibility Project: Cancer Biology |

The Reproducibility Project: Cancer Biology is funded by the Laura and Johan Arnold Foundation, provided to the Center for Open Science in collaboration with Science Exchange. The funder had no role in study design or the decision to submit the work for publication.

### Author contributions

BH, EH, CD, MS, NV, KO, Drafting or revising the article; RP:CB, Conception and design, Drafting or revising the article

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
