## [Decision Letter]

Thank you for submitting your work entitled "Registered report: A chromatin-mediated reversible drug-tolerant state in cancer cell subpopulations" for peer review at *eLife*. Your submission has been favorably evaluated by Sean Morrison (Senior editor), a Reviewing editor, and three reviewers.

The reviewers have discussed the reviews with one another and the Reviewing editor has drafted this decision to help you prepare a revised submission.

For the Reproducibility Project in Cancer Biology, the authors here try to reproduce the critical findings of Sharma et.al. Cell 2010. The authors’ main goal is to establish the reproducibility of generation of drug tolerant persisters (DTP) and characterize them for their growth properties, cell cycle and CD133 expression.

To reproduce the main findings described by Sharma et al., the authors describe important protocols. The authors’ main aims of this project are:

1) To establish reproducibility of generation of DTP;

2) To independently prove reversible nature of DTPs upon drug withdrawal;

3) To elucidate that the observed reversible DTP is linked to epigenetic events marked by Histone acetylation;

4) To examine the effect of histone acetylation inhibitors as a combination target.

This review takes a technical and statistical perspective only. The protocols proposed provided highly detailed experimental conditions, lists of materials and reagents, timings, concentrations and dilutions, data collection methods, and statistical analyses. With such details, the protocols are fully transparent, but some elements of the details are not clearly justified. Several clarifications could be made in the protocols.

Essential revisions:

1) In “Assessing DMSO solvent toxicity on PC9 cells”, two different cell densities (2500 or 5800 cells/well) were presented. Clarification could be made regarding which density should be used in which wells, and to explain the rationale behind having 2 different densities.

2) Along these lines, for experiment 4 under Protocol 1, it might be advisable to try the experiments with the lower density plating also. Since this is a longer term (9 day) experiment, the 2500 cell/well experiment would allow for expansion without crowding.

3) In Protocol 1, the authors intended to determine the growth characteristics of DTP. The protocol is very well described with detailed descriptions of each step required to meet this aim. However, for dose response curve (“Erlotinib dose-response curve on PC9 cells”), is the authors’ intent to do dose response determination by keeping cells with drugs for 9 days? Sharma et al. performed this by incubating the cells with drug for 72hrs only. The authors should use the method to generate IC50 values as described in the original paper.

4) Evaluation of the counting method was not fully described (point 5h in Procedure, Protocol 1: “In order to evaluate a counting method for larger treatments, randomly select counts from 5 dishes (10% of dishes) and calculate the average and standard deviation. Compare these results to the results obtained from all 50 counts”.) What are the evaluation criteria to determine whether sampling 10% of the dishes was enough? Are there any statistical analyses that could quantitatively decide this?

5) Clarification could be made regarding quantitative measures of viability and recovery/growth after detachment (please see point 6e in Procedure, Protocol 1: “Select the detachment method that has the least effect on cell viability immediately after detachment and that allows for cell recovery and growth after seeding into a new plate”). What are the different detachment methods?

6) A standardised set of instructions for manual counting could be added in order to reduce random human error. For instance: counting is performed under 100x magnification, and is repeated 3 times for each visual field.

7) Some materials or procedures were substituted, as reported at the end of each protocol. These changes from the original experiments should be justified. For instance, what are the advantages (or expected changes) of using DMSO-treated cells instead of untreated cells (please see “Known differences from the original study”)?

8) A question on the procedure for protocol 4 is whether acid extracted histones will be examined. From the lysis conditions described in point 3, this is not clear. The inclusion of total H3 as a loading control is a positive.

9) Under Protocol 6, a "known difference for the original study" is noted to be the lack of inclusion of all the pharmacological agents originally reported. The reasoning for this is unclear, and it seems important to include the full panel of HDAC inhibitors utilized in the original study.

10) The authors mention doing STR after each protocol: will it be done for each and every protocol whenever they are using the cells for the experiments?

Statistics:

1) The statistical tests and sample size calculations for each protocol are generally well presented, with clear aims and plans for analysis. However, the protocols and the power calculations would benefit from a bit more justification on why each method of analysis was necessary and how this was related to what was done in the original report. For example, three of the protocols state they will use MANOVA but we cannot see the link to this for the sample size section. What overall effect size was seen in the original report for the MANOVA analyses?

2) It is a shame that the link to the R scripts is not available, as we have no idea what the R code may do and hence cannot judge if it is appropriate.

3) We like the way that the report addresses the unknown error for some of the measurements and hence they plan an interim analysis to re-estimate the standard deviation and increase the sample size if necessary. However, was it not possible to contact the authors of the original paper to get the raw data?

4) We do not understand why the Cochran Mantel Haenszel test will be used (which is a comparison of proportions) if there will be only 3 observations per cell? The simulation states it assumes a normal distribution, and the outcome appears to be a percent reported for each observed unit, rather than a proportion that is a summary statistic for each group (“The estimated data was used to create simulated data sets with preserved sampling structure assuming a normal distribution […] The power was then calculated by counting the number of times p≤0.05 and dividing by 10,000”). We would be concerned that the asymptotic chi-squared distribution property would not hold with so few observations in each cell. If the R code had been viewable here, this may have been clarified.

5) Justify the thresholds at which effect sizes and a priori power are acceptable, and why they were not always the same (for instance, the effect sizes and a priori power were both variable in some tables, different for each group of comparison; “pIFG-1R normalized signal”).

Major comments for Protocol 2 for generation of PC9 DTP:

Dose of erlotinib (please see point 2e in Procedure, Protocol 2): Sharma et al. described use of 2 µM of erlotinib for further experiment, but here authors state to use 100X of IC50. Please use both of these concentrations if these doses turn out to be different.

Major comments for Protocol 3 for survival assay:

1) Generation of drug withdrawn PC9 DTP (please see point 2 in Procedure, Protocol 3). The authors previously calculated doubling time for treatment naïve PC9 and DTP population, but doubling time must be different from drug withdrawn DTP. Hence, the authors should determine the doubling time for drug withdrawn DTPs as well.

2) Cell number for DTP (please see point 3 in Procedure, Protocol 3). The authors describe using 2500 cell/well for DTP and earlier 5800 cells/well for PC9. The authors should either keep the same cell number or give a justification for such difference.

---

## [Author Response]

Essential revisions:

*1) In “Assessing DMSO solvent toxicity on PC9 cells”, two different cell densities (2500 or 5800 cells/well) were presented. Clarification could be made regarding which density should be used in which wells, and to explain the rationale behind having 2 different densities.*

We have included the rationale for the two different seeding densities. To summarize, the cells are used at a low density (2500 cells/well) for IC50 determination (Protocol 3) and at a higher density (5800 cells/well) for DTP generation (Protocol 2).

*2) Along these lines, for experiment 4 under Protocol 1, it might be advisable to try the experiments with the lower density plating also. Since this is a longer term (9 day) experiment, the 2500 cell/well experiment would allow for expansion without crowding.*

Thank you for the suggestion to reflect the timing and density of the survival assays used in the original study; 2,500 cells/well will be used and the experiment will be for 72 hr as indicated in Protocol 3. However, as noted below (Major comment for Protocol 2), we have revised the manuscript and will not perform this preliminary experiment since the dose of erlotinib will not be changed.

*3) In Protocol 1, the authors intended to determine the growth characteristics of DTP. The protocol is very well described with detailed descriptions of each step required to meet this aim. However, for dose response curve (“Erlotinib dose-response curve on PC9 cells”), is the authors’ intent to do dose response determination by keeping cells with drugs for 9 days? Sharma et al. performed this by incubating the cells with drug for 72hrs only. The authors should use the method to generate IC50 values as described in the original paper.*

Thank you for the suggestion to use the method to generate IC50 values as described by Sharma et al. This is how Protocol 3 is outlined. However, as noted below (Major comment for Protocol 2), we have revised the manuscript and will not perform this preliminary experiment since the dose of erlotinib will not be changed.

*4) Evaluation of the counting method was not fully described (point 5h in Procedure, Protocol 1: “In order to evaluate a counting method for larger treatments, randomly select counts from 5 dishes (10% of dishes) and calculate the average and standard deviation. Compare these results to the results obtained from all 50 counts”.) What are the evaluation criteria to determine whether sampling 10% of the dishes was enough? Are there any statistical analyses that could quantitatively decide this?*

We have revised this protocol to reflect a strategy that will utilize an automated cell counter instead of manual counting. While this is dependent also on the detachment method, it more accurately describes how the cells will be counted for each protocol that requires counting the DTPs prior to analysis.

*5) Clarification could be made regarding quantitative measures of viability and recovery/growth after detachment (please see point 6e in Procedure, Protocol 1: “Select the detachment method that has the least effect on cell viability immediately after detachment and that allows for cell recovery and growth after seeding into a new plate”). What are the different detachment methods?*

The different detachment methods are listed under point b of experiment 6 in Protocol 1. Trypsin (at 37˚C and room temperature, and 2-8˚C) and accumax (at room temperature) will be used. Additionally, we have included additional details about the process for determining the detachment method.

*6) A standardised set of instructions for manual counting could be added in order to reduce random human error. For instance: counting is performed under 100x magnification, and is repeated 3 times for each visual field.*

In Protocol 6, the method of counting the number of colonies has been expanded. Regarding the manual counting mentioned in Protocol 1, we have revised the manuscript to utilize an automated cell counter instead to reduce human error.

*7) Some materials or procedures were substituted, as reported at the end of each protocol. These changes from the original experiments should be justified. For instance, what are the advantages (or expected changes) of using DMSO-treated cells instead of untreated cells (“Known differences from the original study”)?*

We have expanded this section of each protocol to clarify the rational when not specified. Regarding, the specific instance of using DMSO vs untreated, we have revised the manuscript to reflect using untreated cells as is originally described. We will also include DMSO-treated cells, the solvent used to dissolve TSA, to understand any impact DMSO treatment has on the viability and cell cycle profile of the cells as an additional exploratory measure.

*8) A question on the procedure for protocol 4 is whether acid extracted histones will be examined. From the lysis conditions described in point 3, this is not clear. The inclusion of total H3 as a loading control is a positive.*

The original paper has no indication of using acid extraction nor did the authors provide any information about this, so the protocol reflects lysis of cells using RIPA buffer.

9) Under Protocol 6, a "known difference for the original study" is noted to be the lack of inclusion of all the pharmacological agents originally reported. The reasoning for this is unclear, and it seems important to include the full panel of HDAC inhibitors utilized in the original study.

We plan to restrict the number of pharmacological agents to the ones utilized in other experiments included in the replication attempt. We agree that the exclusion of certain experiments limits the scope of what can be analyzed by the project, but we are attempting to identify a balance of breadth of sampling for general inference with sensible investment of resources on replication projects to determine to what extent the included experiments are reproducible. As such, we will restrict our analysis to the pharmacological agents being replicated and will not include discussion of those not included in this study.

*10) The authors mention doing STR after each protocol: will it be done for each and every protocol whenever they are using the cells for the experiments?*

The STR profile and mycoplasma test will be performed after the cell line is acquired by the replicating lab and before freezing down the cell line or after a few months of actively maintaining them in culture. This does not mean it will be done for each protocol independently, since most experiments will be performed as close together as possible.

*Statistics: 1) The statistical tests and sample size calculations for each protocol are generally well presented, with clear aims and plans for analysis. However, the protocols and the power calculations would benefit from a bit more justification on why each method of analysis was necessary and how this was related to what was done in the original report. For example, three of the protocols state they will use MANOVA but we cannot see the link to this for the sample size section. What overall effect size was seen in the original report for the MANOVA analyses?*

We have revised the manuscript to include discussion in the power calculation section, and where necessary the analysis plans, to discuss the approach. Specifically we highlighted the additional normalization controls not included in the original study, which are exploratory to the original paper. In regards to the question about the MANOVA analysis, the original paper presented the Western blots as single images. Since we lack raw data are unable to perform the MANOVA analysis and instead are using a 2-way ANOVA to estimate sample size. We have included this in the power calculation section.

*2) It is a shame that the link to the R scripts is not available, as we have no idea what the R code may do and hence cannot judge if it is appropriate.*

There was an error when sharing the private link. The revised manuscript does not have the ‘h’ at the end of the url that caused the error. The scripts and other files should be accessible now at: https://osf.io/q9bxy/?view_only=cb3c377d53fe4d48985025a63da97091

*3) We like the way that the report addresses the unknown error for some of the measurements and hence they plan an interim analysis to re-estimate the standard deviation and increase the sample size if necessary. However, was it not possible to contact the authors of the original paper to get the raw data?*

We did contact the authors and provided feedback on protocol details where possible. In respect to the raw data we were informed they were not accessible.

*4) We do not understand why the Cochran Mantel Haenszel test will be used (which is a comparison of proportions) if there will be only 3 observations per cell? The simulation states it assumes a normal distribution, and the outcome appears to be a percent reported for each observed unit, rather than a proportion that is a summary statistic for each group (“The estimated data was used to create simulated data sets with preserved sampling structure assuming a normal distribution […] The power was then calculated by counting the number of times p≤0.05 and dividing by 10,000”). We would be concerned that the asymptotic chi-squared distribution property would not hold with so few observations in each cell. If the R code had been viewable here, this may have been clarified.*

The R code should now be available using the link above. The reason we used the Cohran-Mantel-Haenszel test is to control for the number of times the experiment is performed since the response variable is influenced by this covariate. Each replicate will have a proportion of cells in each of the three cell cycle phases (G1, S, and G2M) for each of the cell lines (drug-naïve PC9 and PC9-DTPs) being tested against each other. This will test if there is a consistent difference in the proportions across the repeats.

*5) Justify the thresholds at which effect sizes and a priori power are acceptable, and why they were not always the same (for instance, the effect sizes and a priori power were both variable in some tables, different for each group of comparison; “pIFG-1R normalized signal”).*

We have revised the manuscript to discuss how the calculations were done in the instances where only a single biological replicate is reported in the original paper. At each level of variance the effect size was estimated using the values quantified from the reported image as the mean. The estimated effect size at each level of variance was then used to calculate the needed sample size to achieve at least 80% power (with the achieved power reported) and the indicated alpha error. This provides a range of effect sizes and sample sizes for identifying a suitable starting point regarding number of biological replicates. Additionally, the number of replicates will be recalculated after the indicated number of replicates is performed, using the variance from the replication data and the quantified value from the original paper. This will determine if additional replicates are needed.

*Major comments for Protocol 2 for generation of PC9 DTP: Dose of erlotinib (please see point 2e in Procedure, Protocol 2): Sharma et al. described use of 2 µM of erlotinib for further experiment, but here authors state to use 100X of IC50. Please use both of these concentrations if these doses turn out to be different.*

Thank you for highlighting this point. While it would be ideal to perform these experiments both ways (at 2 µM and 100X the replication attempt calculated IC50), it would be an extension of the original work and beyond the scope of this project. As such, the replication attempt will restrict itself to using erlotinib at 2 µM as described in the original paper with the calculated IC50 from the replication attempt (Protocol 3) to assess if 2 µM is over 100X the replication IC50 value. We have thus removed language from the manuscript indicating the concentration of erlotinib is subject to change. The one potential caveat with this approach is the inability to generate DTPs. If during the first protocol, where conditions are optimized, DTPs are not generated or a large fraction of cells survive, we will contact the original authors for advice prior to proceeding with the outlined experiments. If any modifications are made, they will be recorded along with the data justifying the need to make a modification and made available.

*Major comments for Protocol 3 for survival assay: 1) Generation of drug withdrawn PC9 DTP (please see point 2 in Procedure, Protocol 3). The authors previously calculated doubling time for treatment naïve PC9 and DTP population, but doubling time must be different from drug withdrawn DTP. Hence, the authors should determine the doubling time for drug withdrawn DTPs as well.*

The doubling time of drug-withdrawn DTPs is determined in Protocol 1.

2) Cell number for DTP (please see point 3 in Procedure, Protocol 3). The authors describe using 2500 cell/well for DTP and earlier 5800 cells/well for PC9. The authors should either keep the same cell number or give a justification for such difference.

We have included the rationale for the two different seeding densities in Protocol 1. The cells are used at a low density (2500 cells/well) for this protocol, IC50 determination, and at a higher density (5800 cells/well) for DTP generation (Protocol 2).